# LARGE LANGUAGE AND PROTEIN ASSISTANT FOR PROTEIN-PROTEIN INTERACTIONS PREDICTION

## ABSTRACT

Predicting the types and affinities of protein-protein interactions (PPIs) is crucial for understanding biological processes and developing novel therapeutic approaches. While encoding proteins themselves is essential, PPI networks can also provide rich prior knowledge for these predictive tasks. However, existing methods oversimplify the problem of PPI prediction in a semi-supervised manner when utilizing PPI networks, limiting their practical application. Furthermore, how to effectively use the rich prior knowledge of PPI networks for novel proteins not present in the network remains an unexplored issue. Additionally, due to inflexible architectures, existing methods cannot handle complexes containing an flexible number of proteins. To overcome these limitations, we introduce LLaPA (Large Language and Protein Assistant), a multimodal large language model that integrates proteins and PPI networks. LLaPA offers a more rational approach to utilizing PPI networks for PPI prediction and can fully exploit the information of PPI networks for unseen proteins. Through natural language instructions, LLaPA can accept flexible number of protein sequences and has the potential to perform various protein tasks. Experiments show that LLaPA achieves state-of-the-art performance in multi-label PPI type prediction and is capable of predicting the binding affinity between multiple interacting proteins based on sequence data.

## 1 INTRODUCTION

Protein-protein interactions (PPIs) are fundamental to biological processes and critical in drug discovery (Wells & McClendon, 2007; Braun & Gingras, 2012). Traditional high-throughput screening methods, such as yeast two-hybrid screens (Ito et al., 2001) and tandem affinity purification (Gavin et al., 2002), are both expensive and time-consuming. Recently, advancements in deep learning have led to numerous approaches for predicting PPIs. These approaches can be divided into those that utilize PPI networks and those that do not. Methods that do not use PPI networks include DPPI (Hashemifar et al., 2018), DNN-PPI (Li et al., 2018), PIPR (Chen et al., 2019), TAGPPI (Song et al., 2022), and Geo-PPI (Liu et al., 2021). These methods encode proteins individually and then concatenate the features of two proteins for downstream tasks. Specifically, DPPI and DNN-PPI perform binary classification to determine protein binding, while PIPR and TAGPPI also predict multi-label PPI types (mPPI). Geo-PPI focuses on changes in protein binding affinity due to mutations.

Methods based on PPI networks encode not only proteins but also the PPI network. In a PPI network, nodes represent proteins, and edges, often multi-labeled, indicate relationships between them. PPI networks are essential for predicting PPIs, as protein interactions depend on both individual features and their positions within the larger network (Lee, 2023). GNN-PPI (Lv et al., 2021) was the pioneering method leveraging PPI networks, achieving significant improvements in the mPPI task. Subsequent methods, such as SemiGNN-PPI (Zhao et al., 2023), HIGH-PPI (Gao et al., 2023b), and MAPE-PPI (Wu et al., 2024b), built upon GNN-PPI's settings and demonstrated even better performance in mPPI prediction.

Despite significant advancements, ~~current~~ these methods face three critical limitations: **(1) Oversimplified mPPI Task Setting:** Existing methods utilize connection information between unseen proteins in a semi-supervised manner (Kipf & Welling, 2016; Lv et al., 2021; Gao et al., 2023b; Zhao et al., 2023; Wu et al., 2024b), which oversimplifies task difficulty. Current benchmarks separate a portion of the PPI network data as the test set, and the topological information of the test

set is also input into the model. This approach explicitly informs the model of relationships between protein pairs being tested, facilitating information exchange and simplifying PPI prediction. Unlike readily available protein sequences, acquiring connection information between proteins often requires extensive biological experiments and analysis, making this approach impractical for real-world applications. ~~(2) Limitations in Handling Unseen Proteins:~~ (2) Ineffectiveness of PPI Network Information for Unseen Proteins: In real-world scenarios, we frequently encounter unseen proteins that do not exist in any PPI network. Existing methods fail to effectively utilize PPI network information in such cases, as the model cannot extract useful information from the network topology, thereby affecting prediction accuracy and practicality. **(3) Limitations in Multi-Protein Interactions:** ~~Current~~ These models, with their fixed architectures, can only handle interactions between two proteins and cannot predict relationships involving multiple proteins or the affinity of multi-protein complexes. Many biological processes depend on multi-protein complexes, such as antigen-antibody complexes, which typically consist of three chains: the antigen, the heavy chain of the antibody, and the light chain of the antibody (Wu et al., 2024a). The challenge lies in the unknown number of proteins, requiring models to be flexible enough to accept an arbitrary number of proteins as input. ~~Existing~~ These methods struggle with such complex multi-protein interactions, limiting their applicability in practical biological research.

Recently, some studies have achieved notable performance in protein encoding and understanding through joint learning of proteins and natural language, such as ProtLLM (Zhuo et al., 2024), Pro-LLama (Lv et al., 2024), Prot2Text (Abdine et al., 2024), and ProteinGPT (Xiao et al., 2024). Pre-trained on large-scale protein databases, these methods exhibit strong generalization capabilities. The flexibility of LLMs enables them to handle tasks involving multiple protein sequences, addressing Challenge (3) effectively. Nonetheless, they did not further explore the task of multi-sequence proteins, nor did they utilize the rich information provided by the PPI network.

~~To address these challenges~~ In this work, we propose a multimodal model called LLaPA (Large Language and Protein Assistant), which effectively addresses the aforementioned three challenges simultaneously. LLaPA integrates protein representations and PPI networks into a large language model (LLM). We construct a more general PPI network, inputting both network topology information and protein information into the LLM to assist in decision-making. During both training and inference, we completely remove edges that overlap between the PPI network and the test set. Treating the PPI network as external knowledge, we inject this knowledge into the LLM prompt using Retrieval-Augmented Generation (RAG) (Gao et al., 2023a). For proteins not present in the PPI network, we find similar protein nodes within the PPI network and provide their topology as additional information. Leveraging the flexibility of large language models, LLaPA can accept flexible number of proteins as input and use natural language instructions for downstream tasks.

The contributions of this paper can be summarized as follows:

- We reveal the limitations of existing methods in utilizing PPI networks and provide a straightforward method for more reasonable utilization of PPI networks.

- We propose treating the PPI network as external knowledge and injecting it into LLMs through RAG to assist downstream tasks. This approach is also effective for unseen proteins. We also constructed a more general PPI network called UPPIN.

- We develop a protein natural multimodal large language model, LLaPA, which integrates the protein encoder EMS-2 (Lin et al., 2022), the PPI network encoder SGC (Wu et al., 2019), and the large language model llama3-8b (Touvron et al., 2023). LLaPA can handle flexible numbers of proteins and has the potential perform diverse protein tasks.

- Experiments show that LLaPA achieves state-of-the-art (SOTA) performance on the mPPI task and demonstrates significant accuracy in multi-sequence affinity prediction.

## 2 RELATED WORK

### 2.1 PROTEIN-PROTEIN INTERACTIONS

Protein-Protein Interactions (PPIs) are crucial components of cellular activities and play significant roles in various biological functions (Lu et al., 2020; Bryant et al., 2022; Richards et al., 2021). The interactions among multiple proteins form complex PPI networks, which implicitly represent the

signaling processes and pathways of various life activities within organisms. Understanding PPIs not only helps us decipher complex biological systems but also aids in identifying potential targets for disease intervention.

With the rise of deep learning technologies, researchers have proposed numerous deep learning-based methods for PPI prediction. From a task perspective, PPI tasks include: (1) Binary Classification: This task involves inputting a pair of protein sequences and determining whether these two proteins can interact. Methods such as DPPI, DNN-PPI, PIPR, and TAGPPI typically include a convolutional neural network module as the protein encoder. After encoding the two proteins separately, a feature fusion module combines the encoded features, and a binary classifier outputs the classification result. (2) Multi-label PPI Type Prediction: This task focuses on identifying the types of interactions between two proteins. PIPR and TAGPPI can also handle this task. GNN-PPI introduces the topological information of the PPI network, combining the topological information of proteins in the PPI network with protein features, achieving significant improvements in the mPPI task. Subsequent works like HIGH-PPI and MAPE-PPI use the same PPI network. (3) Protein-Protein Binding Affinity Prediction: This task typically focuses on predicting changes in binding affinity between protein complexes due to mutations, as seen in works like Geo-PPI, DDAffinity (Yu et al., 2024), and topNettree (Wang et al., 2020). These methods input the original and mutated protein features to predict the affinity changes caused by specific mutations. Few works directly predict the binding affinity of protein complexes, with PIPR being one known example. These methods can only handle pairwise protein interactions and cannot predict the affinity of multi-sequence complexes. (4) PPI Binding Site Prediction: This task requires amino acid-level encoding. Representative works include DeepHomo (Yan & Huang, 2021), GLINTER (Xie & Xu, 2022), and DeepInter (Lin et al., 2023), which are beyond the scope of this discussion. (5) Protein-Protein Conformation Prediction: Similar to task (4), this also requires amino acid-level encoding and is not covered in this paper.

## 2.2 MULTIMODAL LARGE LANGUAGE MODELS

Multimodal Large Language Models (MLLMs) are dedicated to enabling LLMs to recognize and understand non-natural language modality data, such as images, sounds, etc. ~~From a methodological perspective,~~ A common approach involves first using multimodal encoders to encode data from various modalities. Then, a projector module aligns the output space of these modalities with the input space of the LLM. This process injects multimodal output features into the LLM, enabling it to understand non-natural language modalities. Subsequently, Multimodal Instruction Tuning is employed, which mixes natural language instructions with multimodal data, allowing the MLLM to perform downstream tasks based on the given multimodal data and natural language instructions. Representative works in ~~this area~~ include LLaVA (Liu et al., 2024a), InstructBLIP (Dai et al., 2023), VisionLLM (Wang et al., 2024), and MultiModal-GPT (Gong et al., 2023). ~~These models exemplify the integration of multimodal data into LLMs, enhancing their ability to process and understand diverse types of data.~~ Recently, some efforts have been made to integrate protein modality into LLMs, endowing LLMs with the ability to understand proteins. Relevant work includes ProtLLM (Zhuo et al., 2024), Prot2Text (Abdine et al., 2024), and ProteinGPT (Xiao et al., 2024), which have achieved noteworthy performance in gene ontology term prediction, as well as understanding of protein sequences and structures.

## 3 METHOD

### 3.1 PROBLEM SETTINGS

This work focuses on two tasks: (1) Multi-label PPI type prediction. Given a pair of proteins $(p_1, p_2)$, the goal is to predict the types of interactions between them, which is a multi-class classification task. (2) Multi-sequence Affinity prediction. Given a complex $C = (B, T)$, where $B$ refers to the binder and $T$ refers to the target, predict its logarithmic dissociation constant $\log Kd = \log \frac{[B][T]}{[BT]}$. $B$ and $T$ can each be a single protein sequence or a complex containing multiple sequences. For the PDB2020 (PP) dataset from PDBBind (Liu et al., 2017), which includes 2852 complexes, accurately extracting the binder and target based on the given information is very challenging and requires manual analysis of the papers corresponding to each PDB entry. Therefore, we have simplified this task in the form of: Given a set of proteins $(p_1, p_2, ..., p_k)$, predict its

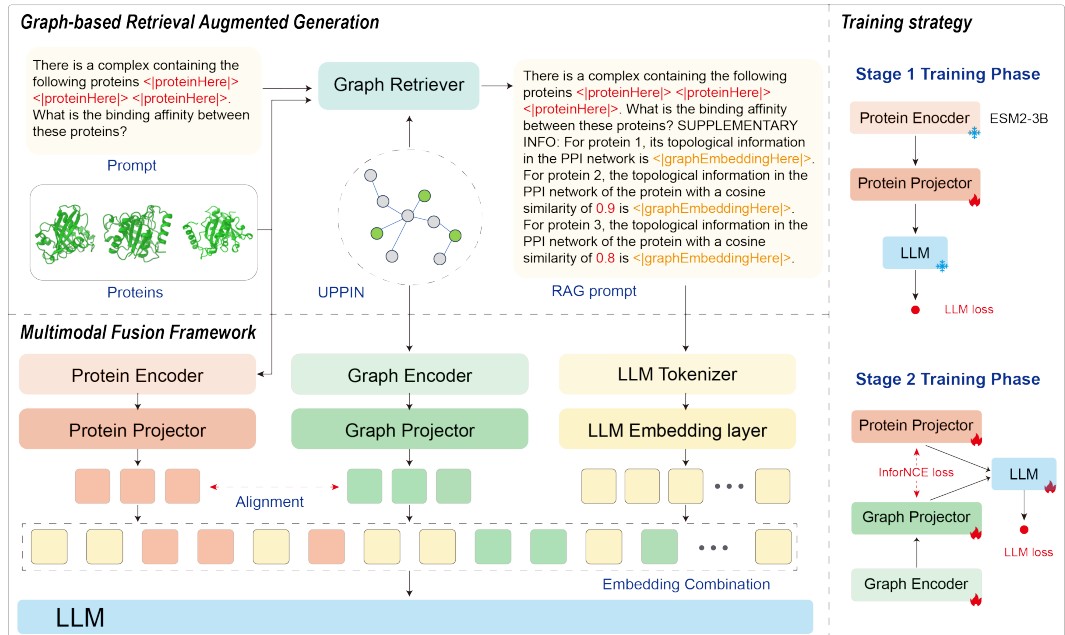

Figure 1: LLaPA is a large language model that integrates proteins and protein-protein interaction (PPI) networks. It consists of two main components: **Graph-based Retrieval Augmented Prompt Preparation** and a **Multimodal Fusion Framework**. In the Graph-based Retrieval Augmented Prompt Preparation stage, we search for matching topological information in UPPIN (a unified PPI network we constructed) based on the input protein and task instructions. This process yields additional topological features and information-enhanced natural language instructions. Subsequently, we input the protein, graph features, and enhanced instructions into our Multimodal Fusion Framework for training and inference. The model training includes two stages: First Stage: We update only the parameters of the protein projector. Second Stage: We simultaneously update the parameters of the protein projector, graph encoder, graph projector, and the LLM. During this stage, we align the output embeddings of the graph projector with the output embeddings of the pre-trained protein projector. This alignment enhances the complementarity of the two modalities and accelerates the alignment of graph output embeddings with the LLM.

$\log Kd$ specified by the dataset. ~~Given a set of proteins $(p_1, p_2, ..., p_k)$, the objective is to predict the logarithmic dissociation constant (log Kd) of the complex formed by these proteins, which is a regression task.~~ We leverage a PPI network to obtain prior knowledge about the target proteins to aid in the prediction. The PPI network is represented as a graph $G = \{V, \mathbf{A}\}$, where $V = \{v_1, v_2, .., v_n\}$ are the nodes of the graph, with each node $v_i$ corresponding to a protein. $\mathbf{A} \in \mathbb{R}^{n \times n}$ is the adjacency matrix of the graph, where $a_{ij} = 1$ if there is an interaction between proteins $p_i$ and $p_j$, and $a_{ij} = 0$ otherwise. We use $\mathbf{X} \in \mathbb{R}^{n \times d}$ to represent the feature matrix of the graph $G$, where each row $x_i$ represents the features of the $i$-th protein.

## 3.2 OVERALL ARCHITECTURE

LLaPA is an integrated large language model that combines protein and graph data, as shown in Figure 1. It consists of two main components: **Graph-based Retrieval Augmented Prompt Preparation** and a **Multimodal Fusion Framework**. In the Graph-based Retrieval Augmented Prompt Preparation stage, we search for matching topological information in the PPI network based on the input proteins. This allows us to obtain additional topological features and enriched natural language instructions. Subsequently, we input the proteins, graph features, and the augmented instructions into our Multimodal Fusion Framework for training and inference. Before formally introducing our method, we will briefly discuss the limitations of existing PPI network-based methods.

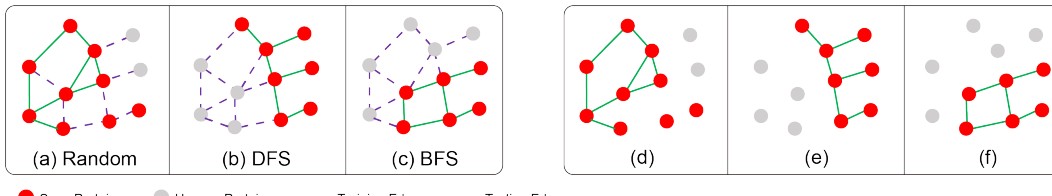

Figure 2: Existing data splitting methods for models based on PPI networks include: (a) Random: randomly selecting a portion of edges from the PPI network as the test set; (b) DFS: using depth-first approach to traverse the PPI network and selecting a portion of edges as the test set; (c) BFS: using breadth-first approach to traverse the PPI network and selecting a portion of edges as the test set. Regardless of the splitting method used, all edges are input into the GNN during the inference process. This allows information exchange between the proteins being tested, greatly simplifying the difficulty of multi-label PPI type prediction and limiting their practical value. We believe that all test edges should be removed during both the training and testing phases, as shown in (d)(e)(f).

### 3.3 LIMITATIONS OF OVERSIMPLIFIED mPPI TASK SETTING

PPI networks are essential for PPI-related tasks, as a protein's position within the network provides valuable prior knowledge. Lv et al. (2021) were the first to apply PPI networks to the mPPI task with their GNN-PPI model, which uses a graph isomorphism network (GIN) (Xu et al., 2018) to encode PPI network topology. They validated the model using three data partitioning methods: random, DFS (depth-first search), and BFS (breadth-first search). As shown in Figure 2-(a), (b), and (c), these methods partition a portion of the edges as the test set, which the labels are not used for training but the structural information are retained for message passing in the GNN model. This allows message exchange between the protein pairs being tested, reducing prediction difficulty. During training, GNN-PPI uses the training edges (green solid edges in Figure 2), but during testing, the test edges (dashed edges in Figure 2) are also included within the graph encoder. Subsequent work, HIGH-PPI, followed GNN-PPI's setup, while MAPE-PPI used all topological information during both training and inference phases.

Unlike protein sequences, which are relatively easy to obtain, acquiring connection information between proteins requires extensive biological experiments and analysis. This makes it challenging to effectively use connection information between test proteins in practical applications. However, removing test set edges during testing may bring another issue, as shown in Figure 2-(d), (e), and (f), where gray nodes become isolated and cannot obtain useful information from the PPI network. Additionally, an unseen protein inherently has no edges in the PPI network, making it an isolated node. Existing methods do not address how to handle this situation.

### 3.4 GRAPH-BASED RETRIEVAL AUGMENTED PROMPT PREPARATION

To address the aforementioned issues, we propose a novel method that utilizes the PPI network as an external knowledge source. By employing the RAG (Retrieval-Augmented Generation) technique, we integrate the knowledge from the PPI network into the input of the large language model (LLM). Given a set of proteins $P = \{p_1, p_2, ..., p_m\}$ and a textual task prompt $W$, we first locate the corresponding nodes $V = \{v_1, v_2, ..., v_m\}$ in the PPI network for each protein. Next, we construct an enhanced natural language instruction $W_{RAG}$. Finally, we input this set of proteins, along with the graph node information and the enhanced instruction into the LLM.

In real-world applications, we may lack prior knowledge about a new protein, meaning it might not exist in any PPI network. In the biological domain, Multiple Sequence Alignment (MSA) (Edgar & Batzoglou, 2006) is a common method for analyzing protein functions. MSA aligns multiple protein sequences to study the structure, function, and evolutionary relationships of the target protein. A key aspect of MSA is the use of reference sequences. Inspired by MSA, for a protein that is an isolated node in the PPI network, we can approximate its topological reference by comparing it with proteins in the PPI network. Specifically, for a protein $p$ not present in the PPI graph $G$, we calculate its similarity to each protein in $G$, denoted as $S = \{s(p, p_i) | p_i \in G\}$. This similarity can be computed using methods such as sequence alignment scores, structural similarity, or other biological metrics;

here, we use the cosine similarity between protein features. With the similarity scores, we retrieve the most similar proteins in the PPI network and use their topological information as a proxy for the isolated protein. This enables us to construct an enriched instruction $W_{RAG}$ that includes relevant topological features, which we then integrate into the LLM prompt for prediction tasks. We did not perform any additional processing on the similarity scores; instead, we directly placed the similarity scores explicitly into $W_{RAG}$.

Our instruction template, shown in APPENDIX Figure 3, uses a structured data representation, separating text, protein sequences, and graph node indices. In the text, we use two special tokens, `<|proteinHere|>` and `<|graphEmbeddingHere|>`, to denote the protein embedding and the PPI network node embedding, respectively. This method can also address isolated nodes caused by the removal of test edges. When constructing $W_{RAG}$, we initially assess the degree of the target protein $p$ within the PPI network. If the degree is 0, indicating that $p$ is an isolated node, we employ the same methodology used for an unseen protein to derive the topological information of similar proteins in the PPI network.

### 3.4.1 UNIFIED PPI NETWORK

Although leveraging protein similarity to utilize the topological information of similar proteins in the PPI network is beneficial, the number and diversity of protein nodes remain a limitation. Intuitively, a PPI network with more proteins and greater diversity enhances the model's generalization ability. To achieve better generalization, we constructed a larger PPI network called UPPIN (Unified PPI Network), which consists of three sub-datasets: STRING (Homo sapiens subset) (Szklarczyk et al., 2016), PDBBind (Liu et al., 2017), and SAbDab (Dunbar et al., 2014).

STRING (Homo sapiens subset) is a multi-source PPI network comprising 15,202 unique proteins and 581,161 unique edges. It includes seven types of protein interactions: activation, binding, catalysis, expression, inhibition, posttranslational modification (ptmod), and reaction. For UPPIN, we retained all nodes and edges from STRING but removed the edge labels.

PDBBind is a database derived from the PDB (Protein Data Bank) (Berman et al., 2000), containing biomolecular complexes with experimentally determined binding affinities. We used the 2020 version of PDBBind, which includes 2,852 protein-protein complexes, totaling 5,711 unique proteins. Due to often incomplete protein sequences in the crystal data, we first obtained the fasta data for each protein from the PDB. We then connected each pair of proteins within a complex with an edge, resulting in a total of 5,978 edges.

SAbDab is an antibody-antigen database that includes complexes and experimental information, and it is continuously updated. We used data up to PDB 8cds, comprising 16,226 complexes and 6,315 unique proteins. As with PDBBind, we first obtained the fasta data for each complex from the Protein Data Bank and then constructed edges between each pair of proteins within a complex.

Finally, we merged these three datasets to create our UPPIN, which includes a total of 26,180 unique proteins and 594,216 unique edges. Detailed information is provided in Appendix Table 6.

By constructing UPPIN, we expanded the coverage and diversity of the PPI network, enhancing the model's generalization capability. This larger-scale PPI network provides richer topological information and better supports prediction tasks for unseen proteins.

### 3.5 MULTIMODAL FUSION FRAMEWORK

As illustrated in Figure 1, we encode the protein, graph, and text separately, then fuse them to form the input for the LLM. For a given protein $p$, we use an encoder $f_p(\cdot)$ to obtain the protein features $\mathbf{Z_p} = f_p(p)$. Similar to LLaVA (Liu et al., 2024a), we use a learnable mapping matrix $\mathbf{W}_p$ to map $\mathbf{Z}_p$ to the embedding tokens $\mathbf{H}_p$ for the LLM:

$$\mathbf{H}_p = \mathbf{W}_p \cdot \mathbf{Z}_p, \ \ with \ \ \mathbf{Z}_p = f_p(p).$$

While sophisticated designs like QFormer (Li et al., 2023), C-Abstractor, and D-Abstractor (Cha et al., 2024) exist for connecting different data modalities with LLMs, recent research suggests that a Linear Projector may be optimal when sufficient computational resources are available (Yao et al., 2024). We chose ESM2-3B (Lin et al., 2022) as the protein encoder, a transformer-based model capable of directly encoding amino acid sequences.

For the PPI graph, we first use a graph encoder $f_v(\cdot)$ to encode it as $\mathbf{X}' = f_v(\mathbf{X})$. We then extract the embedding of the node corresponding to protein $p$ in the graph, denoted as $\mathbf{X}'_p$. A mapping matrix $\mathbf{W}_v$ is used to map $\mathbf{X}'_p$ to the embedding tokens $\mathbf{H}_v$ for the LLM:

$$\mathbf{H}_v = \mathbf{W}_v \cdot \mathbf{X}'_p, \ \ with \ \ \mathbf{X}' = f_v(\mathbf{X}).$$

We use SGC (Wu et al., 2019) as the graph encoder, which balances encoding capability and computational efficiency. The graph is encoded using the following formula:

$$\mathbf{X}' = (\hat{\mathbf{D}}^{\frac{-1}{2}} \hat{\mathbf{A}} \hat{\mathbf{D}}^{\frac{-1}{2}})^K \mathbf{X} \boldsymbol{\Theta},$$

where $\hat{\mathbf{A}} = \mathbf{A} + \mathbf{I}$ denotes the adjacency matrix with inserted self-loops, $\hat{\mathbf{D}}_{ii} = \sum_j \hat{\mathbf{A}}_{ij}$ is the diagonal degree matrix, and $\boldsymbol{\Theta} \in \mathbb{R}^{d \times d'}$ is the weight matrix. The parameter $K$ controls the number of hops or the receptive field of the convolution.

For an input $(P, V, W_{RAG})$, we obtain the protein embeddings $(\mathbf{H}_{p_1}, \mathbf{H}_{p_2}, ..., \mathbf{H}_{p_m})$ and the corresponding graph node embeddings $(\mathbf{H}_{v_1}, \mathbf{H}_{v_2}, ..., \mathbf{H}_{v_m})$ using the methods described above. We then use the encoding layer of the LLM to obtain the corresponding text embeddings $(\mathbf{H}_{w_1}, \mathbf{H}_{w_2}, ..., \mathbf{H}_{w_n})$. Finally, we combine these embeddings into a complete input: $(\mathbf{H}_w, \mathbf{H}_p, \mathbf{H}_v)$, where the order of these embeddings depends on the positions of different types of tokens in $W_{RAG}$.

## 4 MODEL TRAINING

Our training process is divided into two steps. In the first step, we unload the graph encoder and freeze both the LLM and the protein encoder, training only the protein projector $\mathbf{W}_p$ to map the protein features into the LLM's input space. For this step, we use the UniProtQA (Luo et al., 2023) dataset, which contains 569,516 proteins and 1,891,506 protein question-answer pairs. Each QA record includes only one protein sequence, with questions covering protein functions, official names, families, and sub-cellular locations. Given a QA record $(Q, A, p)$, where $Q$ represents the question, $A$ represents the response, and $p$ is a protein sequence, the objective is to maximize the probability $\mathcal{P}(A|Q, p)$. We optimize this probability using the LLM's autoregressive objective:

$$L_{LLM} = - \sum_{t=1}^{T} \log \mathcal{P}(w_t | w_1, ..., w_{t-1}, p)$$

In the second step, we fine-tune directly on the downstream task. Here, we freeze the protein encoder and load the graph encoder. We update the graph encoder $f_v(\cdot)$, the graph mapping matrix $\mathbf{W}_v$, the protein mapping matrix $\mathbf{W}_p$, and the weights of the LLM. Unlike the protein encoder and protein projector, which have been pre-trained, the weights of the graph encoder and graph projector are randomly initialized. Since the topological information provided by the graph complements the corresponding protein information, and the protein feature projector already connects protein features to the LLM, we can leverage this complementarity to accelerate the alignment of graph features with the LLM. We use InfoNCE (Oord et al., 2018) to maximize the mutual information between the protein representation and the corresponding topological information. The objective function is as follows:

$$L_{infoNCE} = -\mathbb{E}[log \frac{exp(E(\mathbf{H}_p, \mathbf{H}_v))}{\sum_{v'} exp(E(\mathbf{H}_p, \mathbf{H}_{v'}))} + log \frac{exp(E(\mathbf{H}_p, \mathbf{H}_v))}{\sum_{p'} exp(E(\mathbf{H}_{p'}, \mathbf{H}_v))}],$$

where $E(\cdot)$ is an energy function, which can be of flexible form. We use the dot product for its simplicity, i.e., $E(\mathbf{H}_p, \mathbf{H}_v) = \mathbf{H}_p \cdot \mathbf{H}_v$. The advantage of this approach is that we do not need to design additional tasks to pre-train the graph encoder and projector. Therefore, the loss function for the second stage of training is:

$$Loss = L_{infoNCE} + L_{LLM}.$$

# 5 EXPERIMENTS

LLaPA can accept flexible number of protein inputs and adapt to various downstream tasks using a unified loss function through generative methods by constructing different natural language instructions. We evaluated LLaPA's capabilities on multi-label PPI type prediction (mPPI) and multi-sequence affinity prediction (MA).

**Datasets.** For the mPPI task, we used SHS27k and SHS148k, two subsets of STRING constructed by Chen et al. (2019). They randomly selected 3,000 and 8,000 proteins with sequence identity less than 40% from STRING to form these datasets. SHS27k and SHS148k contain 26,945 and 148,051 interaction cases, respectively. We adopted the same data splitting algorithms as GNN-PPI: random (randomly selecting test edges from the PPI network), DFS (using depth-first search to obtain test edges), and BFS (using breadth-first search to obtain test edges). Depending on the splitting algorithm, the test protein pairs can be categorized into three modes: BS (both proteins have been seen), ES (either one protein has been seen), and NS (neither one has been seen). For more detailed information, please refer to the GNN-PPI (Lv et al., 2021) and MAPE-PPI (Wu et al., 2024b) papers. We split these two datasets into training, validation, and test sets in a 60%:20%:20% ratio. For the MA task, we used PDB2020 for validation. We split PDB2020 into a training set and a test set at a ratio of 80%:20%.

**Baselines.** For the mPPI task, we compared against DPPI, DNN-PPI, PIPR, ESM2-3B (fixed), ESM2-3B (ft), ProtLLM, GNN-PPI, HIGH-PPI, and MAPE-PPI. Since GNN-PPI, HIGH-PPI, and MAPE-PPI can see the complete PPI network during the test phase, for a fairer comparison, we removed the edges contained in the test set from the PPI network during the test phase, denoted as GNN-PPI/R, HIGH-PPI/R, and MAPE-PPI/R, respectively. For ESM2-3B (fixed), we fixed the parameters of ESM2-3B and trained a multi-classifier on the top of it; For ESM2-3B (ft), in addition to training multi-classifiers, we also fine-tuned all the weights of ESM2-3B, with all training hyperparameters kept consistent with LLaPA; For ProtLLM, we modified the original code provided by the authors to support the mPPI task and fine-tuned it using the pre-trained weights and hyperparameters supplied by the authors. ~~For the MA task, to the best of our knowledge, LLaPA is the first model capable of handling flexible number of sequences, so we did not find suitable baselines. For reference,~~ For the MA task, we used PIPR to predict the binding affinity of all two-protein complexes in the test set. Additionally, we trained three affinity prediction models using ESM2-3B, named E(2), E(3), and E(4). E(2) is used to predict the binding affinity of two-protein complexes, while E(3) predicts the binding affinity of three-protein complexes, and E(4) for four-protein complexes.

Table 1: Experimental results for multi-label PPI type prediction (micro-F1). The performance of methods based on the PPI network drops sharply after removing the edges contained in the test set. LLaPA achieves the best results under various experimental settings. Bold and underline are used to highlight the first and second scores respectively.

| | SHS27k | | | SHS148k | | |
|---|---|---|---|---|---|---|
| | **random** | **dfs** | **bfs** | **random** | **dfs** | **bfs** |
| DPPI | 70.45 | 43.69 | 43.87 | 76.10 | 51.43 | 50.80 |
| DNN-PPI | 75.18 | 48.90 | 51.59 | 85.44 | 56.70 | 54.56 |
| PIPR | 79.59 | 52.19 | 47.13 | 88.81 | 61.38 | 58.57 |
| ESM2-3B (fixed) | 47.58 | 42.50 | 41.97 | 48.92 | 43.06 | 41.25 |
| ESM2-3B (ft) | 79.23 | 63.38 | 48.80 | 87.86 | 66.92 | 61.88 |
| ProtLLM | 48.67 | 42.77 | 41.94 | 49.29 | 42.66 | 40.33 |
| GNN-PPI/R | 40.53 | 43.19 | 42.52 | 39.48 | 40.96 | 41.42 |
| HIGH-PPI/R | 41.51 | 40.06 | 39.87 | 42.81 | 51.06 | 45.94 |
| MAPE-PPI/R | 76.84 | 51.69 | 55.21 | 85.96 | 62.13 | 56.68 |
| LLaPA | **82.49** (+2.90) | **69.54** (+6.16) | **67.21** (+12) | **91.78** (+2.97) | **73.93** (+7.01) | **70.90** (+9.02) |

## 5.1 Multi-label PPI Type Prediction

The experimental results are shown in Table 1. From these results, we can draw two key insights:

**(1) Underperformance of PPI network-Based Methods After Edge Removal.** The PPI network-based methods yield unsatisfactory results after removing the edges contained in the test set. Among the three PPI network-based algorithms, MAPE-PPI/R achieves the best results across various task settings. However, it still falls short of PIPR in several settings. Specifically, on SHS27k (random), MAPE-PPI/R is 2.75 points lower than PIPR; on SHS27k (DFS), it is 0.5 points lower; on SHS148k (random), it is 2.85 points lower; and on SHS148k (BFS), it is 1.89 points lower. The other two PPI network-based models, GNN-PPI/R and HIGH-PPI/R, perform worse than the non-PPI network models across all task settings. These results are expected because graph encoders heavily rely on the graph structure. When the edges in the test set are removed, the graph structure changes significantly, making it difficult for the weights learned on the training set to be effective. This is particularly problematic for the DFS and BFS data splitting methods, which may result in a number of isolated nodes that cannot obtain any useful information during the graph message-passing process. It is worth noting that MAPE-PPI/R still performs relatively well in this scenario, likely due to its pre-training based on VQ-VAE (Van Den Oord et al., 2017), which demonstrates the effectiveness of its proposed micro-environment-based protein encoding.

**(2) Superior Performance of LLaPA.** LLaPA achieves the best results across all task settings. Under the random splitting method, LLaPA outperforms the second-best model by 2.90 on the SHS27k dataset and by 2.97 on the SHS148k dataset. The improvement is modest, which is expected because, under the random method, most proteins in the test set can be well-trained, making the learning task relatively easy. However, the improvements are much more significant under the DFS and BFS methods. Specifically, LLaPA outperforms the second-best model by ~~17.35~~ 6.16 points on SHS27k (DFS), by 12.00 points on SHS27k (BFS), by ~~11.80~~ 7.01 points on SHS148k (DFS), and by ~~12.33~~ 9.02 points on SHS148k (BFS). The performance on SHS148k are better than those on SHS27k, likely because the SHS148k dataset is approximately 5.5 times larger than SHS27k, making it easier for the model to fit the data. We found that training a multi-classifier with fixed ESM2-3B parameters underperformed. While full parameter fine-tuning of ESM2-3B improved results, it still didn't match LLaPA. Despite ProtLLM has similar architectures with LLaPA, it lack of PPI network information led to suboptimal performance on the mPPI task.

These results highlight the robustness and effectiveness of LLaPA, especially in scenarios where the graph structure is altered or when dealing with more challenging data splits.

Table 2: The PDB2020 dataset's division into training and test sets, and the count of complexes with unique sequences.

| Sequence Number | All | Train | Test |
|---|---|---|---|
| 2 | 1857 | 1485 | 372 |
| 3 | 679 | 535 | 144 |
| 4 | 188 | 156 | 32 |
| 5 | 106 | 89 | 17 |
| 6 | 6 | 3 | 3 |
| 7 | 1 | 0 | 1 |
| 9 | 1 | 1 | 0 |
| 13 | 1 | 1 | 0 |
| 14 | 1 | 1 | 0 |
| 16 | 1 | 1 | 0 |
| sum | 2841 | 2272 | 569 |

Table 3: Experimental results of MA prediction on PDB2020, measured by mean absolute error (MAE) and Pearson correlation coefficient (PCC).

| Sequence Number | Methods | MAE ($\downarrow$) | PCC ($\uparrow$) |
|---|---|---|---|
| 2 | PIPR | 1.42 | 0.34 |
| | E(2) | 1.43 | -0.11 |
| | LLaPA | **1.35** | **0.41** |
| 3 | E(3) | 1.24 | 0.13 |
| | LLaPA | **1.11** | **0.51** |
| 4 | E(4) | 1.82 | -0.24 |
| | LLaPA | **1.09** | **0.76** |
| 5 | LLaPA | 1.02 | 0.35 |
| 6 | LLaPA | 2.37 | 0.96 |
| 7 | LLaPA | 0.82 | N/A |
| all | LLaPA | 1.26 | 0.49 |

## 5.2 Multi-sequence Affinity Prediction

The number of unique proteins in the complexes within the PDB2020 dataset ranges from 2 to 16. To better distinguish LLaPA's prediction capabilities for complexes with different numbers of proteins, we grouped these complexes based on the number of unique proteins and evaluated the performance

for each group. The grouping information is shown in Table 2. Most complexes have fewer than 5 unique proteins. Complexes with 7, 9, 13, 14, and 16 unique proteins each have only one instance. As shown in the Table 2, we randomly split the data into training and test sets at an %80:%20 ratio, resulting in no test data for groups 9, 13, 14, and 16. The experimental results are shown in Table 3.

From the results, we can see that, except for group 6, which has a higher MAE, the differences between the other groups are relatively small. To better understand this, we visualized the training and testing data for this group, as shown in APPENDIX Figure 5. In the three tested complexes, the predicted and actual values for DPB:1QFW are identical; for PDB:6ILM, the predicted and actual values differ by 0.9; however, for PDB:3ZIA, the predicted value differs from the actual value by 6.2. The large error for PDB:3ZIA may be due to the fact that, although it has only 6 unique protein sequences, the total number of sequences in the complex reaches 20. The affinity of such a large number of sequences may not be consistent with the distribution of regular complexes. During the training and testing of the model, each unique sequence is input only once, which leads to LLaPA's suboptimal performance in predicting complexes with a large number of identical sequences.

## 5.3 ABLATION STUDY

We conducted ablation experiments to confirm three aspects: (1) whether the pre-trained protein projector is useful; (2) whether the constructed UPPIN network is useful; (3) whether the designed alignment loss function $L_{infoNCE}$ is useful. We conducted experiments on SHS27k, with data partitioned by DFS. As shown in Table 4, using pretraining improved the results by 7.34 compared to not using pretraining, proving that the pre-trained protein projector is an effective alignment method; using UPPIN improved the results by 2.62 compared to using the original PPI network of SHS27k, and by 26.23 compared to not using a PPI network, proving that the constructed UPPIN is effective. This result is also intuitive, as UPPIN introduces more proteins and edges, providing richer topological information, and without using a PPI network is equivalent to encode proteins with parameters fixed ESM-3B; using $L_{infoNCE}$ to align graphs and proteins improved the results by 3.19 compared to not using $L_{infoNCE}$, proving that this alignment method is effective.

Table 4: Ablation experiments on SHS27k using DFS for data partitioning

| Pretrain | PPIs Network | LinfoNCE | F1 |
|:---:|:---:|:---:|:---:|
| ✓ | | ✓ | 43.31 |
| | UNI | ✓ | 62.20 |
| ✓ | OR | ✓ | 66.92 |
| ✓ | UNI | | 66.35 |
| ✓ | UNI | ✓ | **69.54** |

## 5.4 CONCLUSION

We identified and addressed limitation in current multi-label PPI type predictions based on PPI networks. Our solution, a multimodal large language model named LLaPA, treats the PPI network as external knowledge, integrating it into the model via RAG. We innovated a modality alignment method that leverages pre-aligned protein modalities to expedite graph modality alignment. LLaPA not only sets a new standard in multi-label PPI type prediction but also stands as the first model capable of predicting affinities for multi-sequence complexes with flexible number of protein sequences. Beyond this, LLaPA holds promise for a wide range of other protein-related tasks.

### REPRODUCIBILITY STATEMENT

The source code and data will be made publicly available. During the review phase, we have attached the main code of the model. More implementation details can be addressed at APPENDIX.

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

# A APPENDIX

## A.1 IMPLEMENTATION DETAILS AND HYPERPARAMETERS

We finetune the parameters of Llama3-8b using LoRA Hu et al. (2021). The LoRA target modules are `q_proj`, `k_proj`, `v_proj`, `o_proj`, `gate_proj`, `down_proj`, `up_proj`, and `lm_head`. The model is trained using 8 NVIDIA A100 GPUs (80G). Other parameters are detailed in Table 5.

Table 5: Training parameters.

| Parameter | Value |
|---|---|
| lora_alpha | 64 |
| lora_dropout | 0.1 |
| lora_rank | 256 |
| learning_rate | 4e-5 |
| global_batch | 512 |
| lr_scheduler_type | cosine |
| num_warmup_steps | 100 |
| weight_decay | 0.05 |
| max_grad_norm | 0.03 |
| warmup_ratio | 0.03 |
| bf16 | TRUE |

## A.2 DETAILS OF THE CONSTRUCTED UPPIN

Table 6: Information on the constructed UPPIN.

| | nodes | edges |
|---|---|---|
| STRING | 15,202 | 581,161 |
| PDB2020 | 5,711 | 5,978 |
| SabDab | 6,315 | 7,424 |
| sum | 27,228 | 597,563 |
| unique | 26,180 | 594,216 |

## A.3 DETAILS OF MODELS E(2), E(3), E(4)

Models E(2), E(3), and E(4) are all based on ESM2-3B and are used to predict the affinity of complexes consisting of 2, 3, and 4 proteins, respectively. We fixed the parameters of ESM2-3B, encoded

the protein sequences, and concatenated them. Then, we trained the predictors of E(2), E(3), and E(4), each of which is a simple linear layer. The learning rate was set to 5e-4, and the models were trained for 500 epochs. The loss function used was mean square error:

$$L_{MSE} = \sum (y_i - \hat{y}_i)^2.$$

## A.4 Inputs and Outputs Examples

| RAG Prompt for Multi-label PPI type Prediction | RAG Prompt for Multi-sequence Affinity Prediction |
|---|---|
| **Instruction:** There are two proteins, <\|proteinHere\|> and <\|proteinHere\|>. Among the following seven types of relationships (reaction, binding, ptmod, activation, inhibition, catalysis, expression), list all possible relationships between these two proteins. Carefully analyze the given protein features and their corresponding topological information, based on the definition of the seven protein relations, answer this question in the form of 'According to the given protein information, Their relationships include relation(s).' If multiple relationships may exist, separate them with comma. **SUPPLEMENTARY INFO:** For protein 1, the topological information in the PPI network of the protein with a cosine similarity of 1.0 is <\|graphEmbeddingHere\|>. For protein 2, its topological information in the PPI network is <\|graphEmbeddingHere\|>. | **Instruction:** There is a complex containing the following protein sequences <\|proteinHere\|> <\|proteinHere\|> <...>. What is the binding affinity (logKd) between these proteins? Carefully analyze the given protein features and their corresponding topological information, based on the definition of logKd, answer this question in the form of 'Based on the given protein information, the binding affinity of this compound is logKd= [predicted value].' **SUPPLEMENTARY INFO:** For protein 1, the topological information in the PPI network of the protein with a cosine similarity of 1.0 is <\|graphEmbeddingHere\|>. For protein 2, its topological information in the PPI network is <\|graphEmbeddingHere\|>. For protein 3 ...... |
| **Proteins:** [sequence 1, sequence 2] | **Proteins:** [sequence 1, sequence 2, ...] |
| **Protein Indexes:** [index 1, index 2] | **Protein Indexes**: [index 1, index2, ...] |

Figure 3: The input template for Graph-based Retrieval Augmented Prompt. It includes text instructions, protein sequences, and the position of the protein (or similar proteins) in the PPI network. The **Instruction** is encoded by the embedding layer of the LLM, the **Proteins** are encoded by ESM2-3B, and the **Protein Indexes** are used to find the corresponding proteins' positions in UPPIN and obtain the embeddings of these nodes in UPPIN encoded by SGC. Finally, these three parts of the encoding are fused to form the input for the LLM. Please see Section 3.5 for details.

### A.4.1 Example for mPPI task.

**INPUTS:**

**Instruction:** There are two proteins, <|proteinHere|> and <|proteinHere|>. Among the following seven types of relationships (reaction, binding, ptmod, activation, inhibition, catalysis, expression), list all possible relationships between these two proteins. Carefully analyze the given protein features, based on the definition of the seven protein relations, answer this question in the form of 'According to the given protein information, Their relationships include relation(s).' If multiple relationships may exist, separate them with comma. SUPPLEMENTARY INFO: For protein 1, its topological information in the PPI network is <|graphEmbeddingHere|>. For protein 2, its topological information in the PPI network is <|graphEmbeddingHere|>.

**Protein 1:** MGLTVSALFSRIFGKKQMRILMVGLDAAGKTTILYKLKLGEIVTTIPTIGFNVETVEY KNICFTVWDVGGQDKIRPLWRHYFQNTQGLIFVVDSNDRERVQESADELQKMLQEDELRDAVLLVF ANKQDMPNAMPVSELTDKLGLQHLRSRTWYVQATCATQGTGLYDGLDWLSHELSKR

**Protein 2:** MTECFLPPTSSPSEHRRVEHGSGLTRTPSSEEISPTKFPGLYRTGEPSPPHDILHEPP DVVSDDEKDHGKKKGKFKKKEKRTEGYAAFQEDSSGDEAESPSKMKRSKGIHVFKKPSFSKKKEKD FKIKEKPKEEKHKEEKHKEEKKEKKSKDLTAADVVKQWKEKKKKKKPIQEPEVPQIDVPNLKPIFG IPLADAVERTMMYDGIRLPAVFRECIDYVEKYGMKCEGIYRVSGIKSKVDELKAAYDREESTNLED YEPNTVASLLKQYLRDLPENLLTKELMRFEEACGRTTETEKVQEFQRLLKELPECNYLLISWLIVH MDHVIAKELETKMNIQNISIVLSPTVQISNRVLYVFFTHVQELFGNVVLKQVMKPLRWSNMATMPT LPETQAGIKEEMNENEEVINILLAQENEILTEQEELLAMEQFLRRQIASEKEEIERLRAEIAEIQS RQQHGRSETEEYSSESESESEDEEELQIILEDLQRQNEELEIKNNHLNQAIHEEREAIIELRVQLR LLQMQRAKAEQQAQEDEEPEWRGGAVQPPRDGVLEPKAAKEQPKAGKEPAKPSPSRDRKETSI

**PPI network Index:** 11117, 3205

**OUTPUTS:** According to the given protein information, their relationships include activation, inhibition, catalysis.

### A.4.2 EXAMPLE FOR MA TASK.

**INPUTS:**

**Instruction:** There is a complex containing the following proteins `<|proteinHere|>` `<|proteinHere|>` `<|proteinHere|>`. What is the binding affinity (log Kd) between these proteins? Carefully analyze the given protein features, based on the definition of log Kd, answer this question in the form of 'Based on the given protein information, the binding affinity of this compound is log Kd = [predicted value].' SUPPLEMENTARY INFO: For protein 1, the topological information in the PPI network of the protein with a cosine similarity of 0.99 is `<|graphEmbeddingHere|>`. For protein 2, the topological information in the PPI network of the protein with a cosine similarity of 0.8 is `<|graphEmbeddingHere|>`. For protein 3, its topological information in the PPI network is `<|graphEmbeddingHere|>`.

**Protein 1:** MKHHHHHHPMSDYDIPTTENLYFQGAMGTNECLDNNGGCSYVCNDLKIGYECLCPDGF QLVAQRRCEDIDECQDPDTCSQLCVNLEGGYKCQCEEGFQLDPHTKACK

**Protein 2:** SIPWNLERITPPRYRADEYQPPDGGSLVEVYLLDTSIQSDHREIEGRVMVTDFENVPE EDGTRFHRQASKCDSHGTHLAGVVSGRDAGVAKGASMRSLRVLNCQGKGTVSGTLIGLEFIRKSQL VQPVGPLVVLLPLAGGYSRVLNAACQRLARAGVVLVTAAGNFRDDACLYSPASAPEVITVGATNAQ DQPVTLGTLGTNFGRCVDLFAPGEDIIGASSDCSTCFVSQSGTSQAAAHVAGIAAMMLSAEPELTL AELRQRLIHFSAKDVINEAWFPEDQRVLTPNLVAALPPSTHGAAGTAAASHHHHHH

**Protein 3:** MKGSKGSKGSKPMSAEAPEHGTTATFHRCAKDPWRLPGTYVVVLKEETHLSQSERTAR RLQAQAARRGYLTKILHVFHGLLPGFLVKMSGDLLELALKLPHVDYIEEDSSVFAQ

**PPI network Index:** 6091, 7833, 1781

**OUTPUTS:** Based on the given protein information, the binding affinity of this compound is log Kd = 8.7.

### A.5 DETAILED EXPERIMENTAL DATA FOR MA TASK

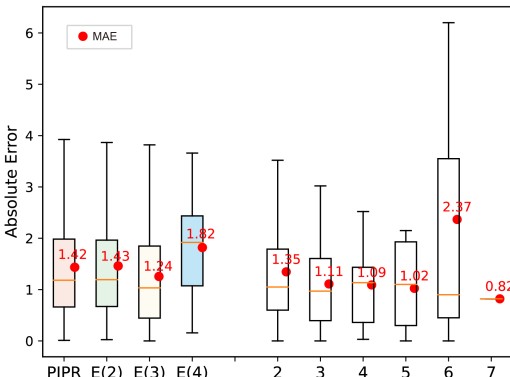

Figure 4: Detailed experimental results for MA task on PDB2020.

### A.6 MORE RELATED WORK: INJECTING GRAPHS TO LARGE LANGUAGE MODEL

Graph data structures have a wide range of real-world applications, such as in financial networks, social networks, and protein interaction networks. When it comes to modeling complex network relationships, graph data structures offer unique advantages. Graph-structured data can also be considered a type of multimodal data, and the approach of injecting multimodal features into LLMs is equally applicable to graph data. For example: GraphLLM (Chai et al., 2023) uses Prefix-tuning (Li & Liang, 2021) to prepend graph features to the input of LLaMA (Touvron et al., 2023). GIT-Mol (Liu et al., 2024b) and Molca (Liu et al., 2023): These models use Q-former (Li et al., 2023) to align graph features with LLMs. The Q-former helps in bridging the gap between graph representations and the input space of LLMs. InstructMol (Cao et al., 2023) employs a mapping matrix $W$ to map

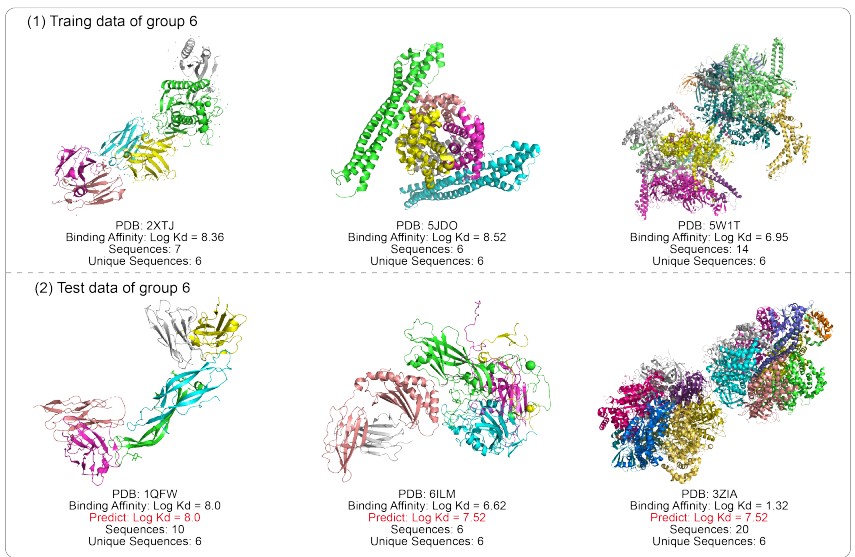

Figure 5: The training data and prediction results of the complex containing 6 unique sequences.

the encoded features of molecular graphs into the input space of LLMs. GraphTranslator (Zhang et al., 2024) uses a Transformer-based Translator Module (Vaswani, 2017) to convert node features of the graph into learnable token embeddings. These methods demonstrate how graph data can be integrated into LLMs, leveraging the strengths of both graph structures and large language models to handle complex, multimodal information. This integration not only improves the models' ability to understand and process graph data but also extends their applicability to a broader range of tasks involving networked data.

## A.7 MORE DETAILS ON THE COMPARISON METHOD BASED ON PPI NETWORK

Table 7: Comparison of PPI network-based methods on the mPPI task with and without removing test edges.

|  | SHS27k | | | SHS148k | | |
| --- | --- | --- | --- | --- | --- | --- |
|  | **random** | **dfs** | **bfs** | **random** | **dfs** | **bfs** |
| GNN-PPI | 87.81 | 71.66 | 66.98 | 90.48 | 76.81 | 71.78 |
| GNN-PPI/R | 40.53 | 43.19 | 42.52 | 39.48 | 40.96 | 41.42 |
|  | *47.28↓* | *28.47↓* | *24.46↓* | *51.00↓* | *35.85* | *30.36↓* |
| HIGH-PPI | 76.62 | 71.69 | 66.75 | 72.21 | 77.32 | 60.08 |
| HIGH-PPI/R | 41.51 | 40.06 | 39.87 | 42.81 | 51.06 | 35.94 |
|  | *35.11↓* | *31.63↓* | *26.88↓* | *29.40↓* | *26.26↓* | *24.14↓* |
| MAPE-PPI | 88.91 | 71.98 | 70.38 | 92.87 | 79.10 | 74.29 |
| MAPE-PPI/R | 76.84 | 51.69 | 55.21 | 85.96 | 61.45 | 56.68 |
|  | *12.07↓* | *20.29↓* | *15.17↓* | *6.91↓* | *17.65↓* | *17.61↓* |

## A.8 LIMITATIONS AND FUTURE WORKS

As a large model integrating proteins, PPI networks, and natural language, LLaPA utilizes natural language instructions and a unified training method for downstream tasks. It can accept a flexible number of protein inputs and has the potential to handle more complex protein tasks. However, LLaPA focuses on protein-level features and is ineffective for tasks requiring amino acid-level features, such as PPI binding site prediction and PPI conformation prediction. Additionally, since

we directly input protein embeddings into the LLM, we cannot leverage the textual features corresponding to protein entities. This is not an issue for novel proteins, but for well-studied proteins with existing literature, utilizing these resources for better analysis is crucial. Furthermore, constructing a larger and more diverse UPPIN is also very important. Therefore, our future work may focus on three aspects: (1) using LLaPA for more comprehensive and challenging protein tasks; (2) incorporating amino acid-level features into LLaPA; (3) enabling LLaPA to better utilize literature resources; (4) constructing a more comprehensive and diverse UPPIN network.

## A.9    ADDITIONAL INFORMATION ON EXPERIMENTAL DATA

Table 8: Proportions of BS (Both have been Seen), ES (Either one protein has been Seen), and NS (Neither one has been Seen) under Different Data Splits.

|       | SHS27k | | | SHS148k | | |
|-------|--------|-----|-----|---------|-----|-----|
|       | random | dfs | bfs | random | dfs | bfs |
| BS | 90.07% | 0.00% | 0.00% | 95.65% | 0.28% | 0.00% |
| ES | 9.28% | 80.13% | 69.54% | 4.26% | 85.62% | 78.10% |
| NS | 0.65% | 19.87% | 30.46% | 0.09% | 14.10% | 21.90% |

## A.10    ADDITIONAL EXPERIMENTS ON THE MA TASK

We conducted additional experiments by modifying the MA setup, inputting all protein sequences from a PDB instead of unique protein sequences. The results of these experiments are shown in 9.

Table 9: Experimental results of inputting all sequences of the complex in the MA task.

| Sequence Number | Train Number | Test Number | MAE | PCCs |
|:---:|:---:|:---:|:---:|:---:|
| 2 | 603 | 150 | 0.7 | 0.69 |
| 3 | 195 | 58 | 0.81 | 0.73 |
| 4 | 513 | 115 | 0.77 | 0.75 |
| 5 | 136 | 27 | 1.04 | 0.52 |
| 6 | 280 | 83 | 1.04 | 0.47 |
| 7 | 30 | 8 | 1.17 | 0.58 |
| 8 | 161 | 42 | 1.04 | 0.63 |
| 9 | 32 | 9 | 1.08 | 0.8 |
| 10 | 80 | 20 | 0.74 | 0.91 |
| 11 | 27 | 4 | 1.26 | 0.57 |
| 12 | 91 | 20 | 0.97 | 0.51 |
| 13 | 2 | 0 | | |
| 14 | 21 | 5 | 1.9 | 0.67 |
| 15 | 9 | 2 | 2.08 | |
| 16 | 19 | 4 | 1.02 | -0.32 |
| 17 | 10 | 0 | | |
| 18 | 21 | 3 | 1.05 | 0.84 |
| 19 | 0 | 1 | 0 | |
| 20 | 7 | 4 | 1.95 | 0.5 |
| 21 | 7 | 3 | 0.62 | 0.92 |
| 22 | 4 | 2 | 0.68 | |
| 23 | 3 | 2 | 0.19 | |
| 24 | 11 | 2 | 1.29 | |
| 26 | 1 | 1 | 0.24 | |
| 27 | 1 | 0 | | |
| 28 | 1 | 0 | | |
| 29 | 1 | 0 | | |
| 30 | 1 | 0 | | |
| 37 | 1 | 0 | | |
| 45 | 1 | 1 | 1.29 | |
| 48 | 1 | 0 | | |
| 54 | 1 | 0 | | |
| 55 | 1 | 0 | | |
| 59 | 1 | 0 | | |
| 63 | 1 | 0 | | |
| 72 | 1 | 0 | | |

