# OpenReview forum: "Large Language and Protein Assistant for Protein-Protein Interactions prediction"
_ICLR.cc/2025/Conference — ICLR 2025 Conference Withdrawn Submission_

### Official Review · Reviewer_7xKG · 2024-11-01

**Soundness:** 3
**Presentation:** 3
**Contribution:** 3
**Rating:** 6
**Confidence:** 4

**Summary:**

The authors propose LLaPA, a multimodal model for predicting protein-protein interactions (PPI) and the binding affinity of protein complexes with a variable number of proteins. Their main contributions are 1) characterization of the performance drop of existing methods using PPI network information when topological information on the test set is restricted, 2) handling PPI networks as external knowledge that can be provided to an LLM via retrieval augmented generation, 3) a large, general PPI network called UPPIN that combines the information from three smaller datasets, and 4) a multimodal approach which incorporates protein features, graph features, and natural language text-based instructions. The authors both compare LLaPA's ability to predict multi-label PPI types to existing methods and evaluate LLaPA's capability to predict multi-sequence binding affinity. LLaPA achieves strong performance, and the authors perform ablation experiments and use multiple baselines.

**Strengths:**

1. The paper is written very clearly and does a good job identifying limitations in existing approaches.
2. There is a strong focus on making LLaPa practical for plausible real-world scenarios, where, for example, a protein sequence that is not a member of an existing PPI network is seen or the goal is to predict the binding affinity on a complex with more than 2 proteins
3. The authors address a major issue of data leakage in protein ML benchmarks, where overlaps between training and test data can inflate performance metrics. To ensure robust evaluation, they completely remove any edges that overlap between the PPI network in the training set and the test set. This approach contrasts with prior evaluations, where such overlaps were not consistently removed, and the authors demonstrate how performances differ for existing methods under this stricter evaluation.
4. The authors clearly demonstrate the effect of using LLaPA to predict multi-sequence binding affinity against the baselines they implement (E(2), E(3), and E(4)), especially since no prior baseline for the 3 and 4 sequences cases existed.
5. The demonstrated performance increase for multi-label PPI class prediction (for DFS and BFS especially) is very strong
6. LLaPA performs pretty well and very stably for various numbers of unique proteins per complex, and the authors investigate the single case where there is unexpectedly high error, providing a plausible explanation for this behavior

**Weaknesses:**

I would appreciate additional details for RAG prompt preparation (addressed in more details in the Questions section), but my primary concern with the paper is that of data leakage for the evaluation:

The authors take steps to ensure that an "unseen protein" is actually unseen with respect to the PPI network (UPPIN), but this doesn't mean that the sequence is unseen with respect to the entire model. Ideally, a protein in the test set should not be seen by any component of the model until inference.
- I understand that this may be difficult and impractical to enforce with large pre-trained protein language models like ESM-2, but it is worth nothing that if ESM-2 has seen two closely related homologs X and Y during training, and LLaPA sees X with additional graph and annotation information (from UniProtQA), then if LLaPA sees sequence Y during inference, high performance can be achieved by simply copying the annotations/predictions for X.
- In the first step of training, the protein projector is trained using UniProtQA, which has function, name, family, and sub-cellular location information about ~600K proteins. I imagine that significant leakage could occur here if there is overlap between the test sets and UniProtQA, since these pieces of information can be very relevant to determining protein complex membership.

These issues of "leakage" may also impact the comparisons against existing methods, which depending on how they are trained, may not have access to all of this information. At the very least, I think it would be worth adding some information about the degree of overlap between UniProtQA and the test sets. It may also be worthwhile to evaluate on a subset of train that is separated from the sequences in UniProtQA by some percent identity threshold at the sequence level.

These may be difficult things to address and should probably be mentioned in the limitations at least.

**Questions:**

1. In Section 3.4 lines 258-260, it is mentioned that the cosine similarity between protein embedding are used to retrieve the most similar proteins in the PPI network. What are the retrieval criteria exactly? Is there a score-based threshold or maybe a limit on the number of similar proteins that can be retrieved?
- There are several works that address predicting sequence similarity/homology at the sequence level using ESM-2 and cosine similarity. It may be worth checking out PLMSearch (Liu et al; 2024) as one example of this.
- Additionally, it may be worth checking out other protein database sequence similarity search tools like HMMER (Accelerated profile HMM searches (Eddy; 2011)), since UPINN only contains ~30K unique sequences. This would be very fast (probably faster than doing comparing the unseen sequence to all nodes in the graph) and could be used to provide more fine-grained control over which sequences are returned (considering where within a sequence matching regions are identified or the domain architecture of a returned sequence).

2. For multi-sequence affinity prediction, why is the sequence number determined by the number of unique sequences? Is is something to do with the the way the prompt is formulated for the LLM? It would be interesting to call sequence number based on the total number of sequences rather than the number of unique sequences, and it may result in a performance boost for LLaPA.

---

> ### Author Response · Authors · 2024-11-22
> **Response to Reviewer 7xKG (1/2)**
>
> Thank you for your recognition of our work, and for your valuable suggestions. We will respond to your questions one by one.
>
> > These issues of "leakage" may also impact the comparisons against existing methods, which depending on how they are trained, may not have access to all of this information.
>
> We have considered this issue. Specifically, UniprotQA has 1,593 duplicate sequences with SHS27k and 4,781 duplicate sequences with SHS148k. We removed the QA data for these sequences during pre-training, so there is no data leakage. The only potential source of data leakage is the protein encoder ESM-3B, as it was trained on UR50, which includes 48 million protein sequences. We added ESM2-3B as a control group, fixed its parameters, and trained a multi-classifier for experiments on the mPPI task. The results show that without fine-tuning the parameters of ESM2-3B, the model struggles to achieve ideal performance. In contrast to the superior performance of LLaPA, this suggests that any potential bias from data leakage in ESM2-3B is minimal.
>
> |            | SHS27k       | SHS27k        | SHS27k     | SHS148k      | SHS148k       | SHS148k       |
> | ---------- | ------------ | ------------- | ---------- | ------------ | ------------- | ------------- |
> |            | random       | dfs           | bfs        | random       | dfs           | bfs           |
> | DPPI       | 70.45        | 43.69         | 43.87      | 76.10        | 51.43         | 50.80         |
> | DNN-PPI    | 75.18        | 48.90         | 51.59      | 85.44        | 56.70         | 54.56         |
> | PIPR       | 79.59        | 52.19         | 47.13      | 88.81        | 61.38         | 58.57         |
> | ESM2-3B    | 47.58        | 42.50         | 41.97      | 48.92        | 43.06         | 41.25         |
> | ProtLLM    | 48.67        | 42.77         | 41.94      | 49.29        | 42.66         | 40.33         |
> | GNN-PPI/R  | 40.53        | 43.19         | 42.52      | 39.48        | 40.96         | 41.42         |
> | HIGH-PPI/R | 41.51        | 40.06         | 39.87      | 42.81        | 51.06         | 45.94         |
> | MAPE-PPI/R | 76.84        | 51.69         | 55.21      | 85.96        | 62.13         | 56.68         |
> | LLaPA      | 82.49(+2.90) | 69.54(+17.35) | 67.21(+12) | 91.78(+2.97) | 73.93(+11.80) | 70.90(+12.33) |
>
>
>
> > In Section 3.4 lines 258-260, it is mentioned that the cosine similarity between protein embedding are used to retrieve the most similar proteins in the PPI network. What are the retrieval criteria exactly? Is there a score-based threshold or maybe a limit on the number of similar proteins that can be retrieved?
> >
> > There are several works that address predicting sequence similarity/homology at the sequence level using ESM-2 and cosine similarity. It may be worth checking out PLMSearch (Liu et al; 2024) as one example of this.
> >
> > Additionally, it may be worth checking out other protein database sequence similarity search tools like HMMER (Accelerated profile HMM searches (Eddy; 2011)), since UPINN only contains ~30K unique sequences. This would be very fast (probably faster than doing comparing the unseen sequence to all nodes in the graph) and could be used to provide more fine-grained control over which sequences are returned (considering where within a sequence matching regions are identified or the domain architecture of a returned sequence).
>
> We initialize the node features of UPPIN using ESM2, and then select the node with the highest cosine similarity based on the target protein p*p* and the features of each protein in UPPIN, obtaining its topological information. Instead of using a threshold, we directly add the obtained topological information and the corresponding cosine similarity value as supplementary information to the prompt.
>
> Thank you for providing information about the efficient sequence comparison tool. We believe that considering the matching regions or domain architecture within sequences could provide a more interpretable and expressive model, which might be significantly beneficial. We will carefully consider your suggestions and aim to improve LLaPA in future work, striving to ultimately build a general, practical, and highly interpretable multimodal protein natural language model.

---

> > ### Author Response · Authors · 2024-11-22
> > **Response to Reviewer 7xKG (2/2)**
> >
> > > For multi-sequence affinity prediction, why is the sequence number determined by the number of unique sequences? Is is something to do with the the way the prompt is formulated for the LLM? It would be interesting to call sequence number based on the total number of sequences rather than the number of unique sequences, and it may result in a performance boost for LLaPA.
> >
> > In our initial version, we input unique sequences to improve computational efficiency, as some complexes in PDB2020 contain up to 40 sequences. This approach is unrelated to the way prompts are created. We believe your suggestion is excellent, and based on it, we conducted additional experiments by directly inputting all sequences from the complexes into the model. Some of the experimental results are as follows:
> >
> > | Sequence  Number | Train Number | Test Number | MAE  | PCCs |
> > | ---------------- | ------------ | ----------- | ---- | ---- |
> > | 2                | 613          | 140         | 1.12 | 0.64 |
> > | 3                | 197          | 56          | 1.16 | 0.51 |
> > | 4                | 502          | 126         | 1.33 | 0.40 |
> > | 5                | 125          | 38          | 1.00 | 0.76 |
> > | 6                | 285          | 78          | 1.42 | 0.50 |
> > | 7                | 29           | 9           | 0.62 | 0.82 |
> > | 8                | 160          | 43          | 1.35 | 0.76 |
> > | 9                | 35           | 6           | 1.21 | 0.72 |
> > | 10               | 81           | 19          | 1.15 | 0.82 |
> > | 11               | 24           | 7           | 1.31 | 0.31 |
> > | 12               | 89           | 22          | 1.13 | 0.64 |
> > | 14               | 22           | 4           | 0.75 | 0.98 |
> > | 18               | 20           | 4           | 1.55 | 0.78 |
> > | 24               | 9            | 4           | 1.76 | 0.26 |
> >
> > Thank you again for your recognition of our work and for your valuable suggestions. We deeply understand that this is not a perfect job, there are many aspects worth improving, and we will continue to refine this work.

---

> > > ### Comment · Reviewer_7xKG · 2024-11-25
> > >
> > > Thank you for adding this experiment. I was curious to see if the MAE for 6 sequences would be reduced (it is) when counting all sequences -- I think these are impressive results demonstrating how LLaPA handles a flexible number of sequences in the MA prediction task

---

> > > > ### Author Response · Authors · 2024-11-27
> > > >
> > > > Thank you for your question. Due to the consideration of all sequences, the grouping has changed. In the initial version, Group 6 had only 6 samples, whereas in the current version, Group 6 has 363 samples. Therefore, a direct comparison with Group 6 from the initial version seems to be of limited significance. We offer two alternative comparison methods:
> > > >
> > > > (1) Average MAE: The average MAE after considering all sequences is 0.87, which is an improvement of 0.39 compared to the initial version's 1.26;
> > > >
> > > > (2) The prediction results for PDB 1QFW, 6ILM, and 3ZIA from the original Group 6 in the new scheme are 7.6, 6.62, and 6.96, respectively (with ground truths of 8.0, 6.62, and 1.32). Among these, 1QFW now belongs to Group 10, 6ILM remains in Group 6, and 3ZIA belongs to Group 20. It can be observed that the prediction for 1QFW has worsened, the prediction for 6ILM has improved, and the prediction for 3ZIA remains poor. Group 20, to which 3ZIA belongs, has 4 samples in total. In our data split (the same as the initial version), none of these 4 samples were included in the training set, which might be one reason for the poor prediction of 3ZIA. Group 20's MAE is 1.94, and PCC is 0.50.
> > > >
> > > > The data we provided in our previous response did not include Group 20 due to a bug in parsing the output results. Thank you again for your question. If you have any further questions, please feel free to ask. Since the discussion has been extended, we have the opportunity to delve deeper into the topic.

---

> > ### Comment · Reviewer_7xKG · 2024-11-25
> >
> > Thank you for your clarifications on my questions and adding the results of this additional experiment! I would just add that data leakage occurs in protein datasets when train and test sets are not split by percent identity, since sequences in train and test can share similar subsequences or can have high sequence-level similarity, which can lead to inflated performance metrics. However, as you stated, the experiment with ESM-2 3B as a control does yield significantly lower performance. My comment was more geared towards generalizability, which is hard to gauge when train and test sequences could be samples from the same underlying distribution of protein sequences.

---

> ### Author Response · Authors · 2024-11-27
>
> Dear Reviewer, we have submitted the rebuttal version of our manuscript. For ease of reading, we have marked the revised content in red and used strikethroughs for the deleted content. We have temporarily moved Figure 3 and Figure 5 to the appendix to create more space.
>
> We greatly appreciate the valuable time and effort you have invested in evaluating this work, your support for our work, and the insightful suggestions you have provided. We agree that considering matching regions or domain architecture within sequences could provide a more interpretable and expressive model. However, this requires more refined data processing and model training, with many details to consider, which we are unable to achieve in the current version. LLaPA is just a beginning, not an end. We believe that there is still a lot of room for exploration and potential research value in integrating protein features with LLMs, and we will carefully consider your suggestions in our future work.
>
> Regarding the MA task you are interested in, we have placed the latest experimental results in Appendix A.10 of the revised version. The data we previously responded to you with had some omissions and inaccuracies due to parsing issues.
>
> Once again, thank you for your trust and support for our work.

---

### Official Review · Reviewer_hDfK · 2024-11-02

**Soundness:** 3
**Presentation:** 3
**Contribution:** 2
**Rating:** 5
**Confidence:** 4

**Summary:**

This paper presents LLaPA (Large Language and Protein Assistant), a multimodal language model aimed at predicting protein-protein interactions (PPIs). Unlike prior graph-based and sequence-based models, which oversimplify PPI networks or struggle with unseen proteins and multi-protein complexes, LLaPA incorporates PPI networks as external knowledge, integrating protein and network data through Retrieval-Augmented Generation (RAG). LLaPA's flexible input allows it to handle varying protein numbers and conduct multiple protein tasks, showing good results in multi-label PPI type prediction and multi-sequence affinity prediction.

**Strengths:**

S1. **Multi-modality Input Handling**: The use of RAG integrates both graph, language, and protein information into the input.

S2. **Good Results**: LLaPA outperforms the explored baselines in PPI type prediction, demonstrating robustness in complex PPI settings and validation tasks.

**Weaknesses:**

W1. **Lack of Comparative Discussion and Limited Contribution**: The most critical issue of the paper is that it ignores relevant protein LLM works which obviously can address the proposed challenges, i.e. relying on structural input and handling multiple protein input. To illustrate, ProtLLM should be discussed and compared. It models a joint space of protein and language, which directly addresses the proposed challenges. Also, there is another work ProLLaMA that does the similar thing. These two works are neither discussed in related work nor are they compared in the experiments.

W2. **Insufficient Benchmarking**: The experiments lack baselines based on protein language models. Specifically, as noted in the original ESM paper, embeddings for protein complexes can be obtained by concatenating the protein chains as input. Given that LLaPA uses ESM embeddings, fine-tuning ESM2 with concatenated inputs is an essential baseline. Additionally, comparisons with ProteinLLM baselines, such as those mentioned in W1, should also be included for a more comprehensive evaluation.

W3. **Redundant Challenges**: Challenges 1 and 2 overlap, both emphasizing issues in graph-based inference at test phase, reducing clarity in presenting the model’s motivation.


References:
- [ProtLLM](https://protllm.github.io/project/), in ACL 24.
- [ProLLaMA](https://arxiv.org/html/2402.16445v1), Arxiv Preprint 2024.

**Questions:**

NA

---

> ### Author Response · Authors · 2024-11-22
> **Response to Reviewer hDfK (1/2)**
>
> Thank you for your recognition of our model framework and experimental results, we also understand your concerns, and we will respond to your concerns one by one.
>
> > W1. **Lack of Comparative Discussion and Limited Contribution**: The most critical issue of the paper is that it ignores relevant protein LLM works which obviously can address the proposed challenges, i.e. relying on structural input and handling multiple protein input. To illustrate, ProtLLM should be discussed and compared. It models a joint space of protein and language, which directly addresses the proposed challenges. Also, there is another work ProLLaMA that does the similar thing. These two works are neither discussed in related work nor are they compared in the experiments.
>
> **Contributions:** Our main contributions are as follows:
>
> 1. We have revealed a significant shortcoming of the existing Multi-label PPI type prediction (mPPI) works and proposed our solution to push the mPPI task towards a more practical direction.
> 2. We propose treating the PPI network as external knowledge and injecting it into LLMs through Retrieval-Augmented Generation (RAG) to assist downstream tasks. We also constructed a more general PPI network called UPPIN.
> 3. We have proposed a multimodal large language model (MLLM) that integrates protein features and PPI network information, achieving state-of-the-art performance in mPPI tasks and also showing promising results in Affinity prediction tasks (MA) tasks.
>
> **More Related Works:** Since LLaPA is a MLLM for proteins, we agree that citing and discussing some works that integrate large language models for proteins can enhance the completeness of this work. Therefore, we will add a section introducing MLLMs for proteins, including related works such as ProtLLM[1] and Prollama[2]. Although both works utilize proteins and LLMs, LLaPA has significant differences from them in terms of model architecture and specific tasks. We will place the detailed discussion in the last paragraph of this response and include it in the revised version of the manuscript.
>
> **Challenges:** The two methods you mentioned do not fully address the challenges we have proposed. We have identified three main challenges:
>
> 1. Existing methods have some unreasonable aspects when utilizing PPI networks, specifically using the topological information between the tested protein pairs during testing, which makes their high performance difficult to apply in real-world scenarios. ProtLLM and ProLLama do not use PPI network information and do not focus on mPPI task, so they cannot address this challenge.
> 2. Handling unseen proteins. We obtain topological information from the PPI network based on sequence alignment concepts, and inject the topological information of the PPI network into the LLM through RAG. ProtLLM and ProLLama do not address this issue and do not provide a solution.
> 3. Predicting relationships between multiple protein complexes. Although these two works do not discuss this issue, from a model architecture perspective, ProtLLM can indeed handle multiple protein sequences. ProLLama, while limited by its context length, can also process a certain number of protein sequences. We will revise this section accordingly.
>
> **Differences between LLaPA, ProtLLM, and ProLLaMA**
>
> 1. ProtLLM designed a pre-training method that interleaves protein and natural language, and validated its effectiveness on several downstream tasks. Although it evaluated on a simple binary classification PPI task, it did not address more challenging mPPI tasks or MA tasks. In these tasks, relying solely on protein features is insufficient for achieving ideal performance; the position of the protein within the PPI network is also crucial. Our work focuses on using LLMs to solve more practically significant problems. Our pre-training strategy is distinctly different from ProtLLM. Additionally, we introduce PPI network features to better predict mPPIs tasks and MA tasks. We also propose leveraging the complementarity of protein information and PPI network information to accelerate the alignment of PPI network features with large natural language model features.
>
> 2. ProLLama focuses on how to effectively train protein sequences directly using large language models. It performed an additional pre-training and an Instruction Tuning based on LLaMA. Besides having a significantly different model architecture from our work, it also did not address mPPI tasks or MA tasks, nor did it utilize PPI network information.
>
> [1] Zhuo, Le, et al. "Protllm: An interleaved protein-language llm with protein-as-word pre-training." *arXiv preprint arXiv:2403.07920* (2024).
>
> [2] Lv, Liuzhenghao, et al. "Prollama: A protein large language model for multi-task protein language processing." *arXiv e-prints* (2024): arXiv-2402.

---

> > ### Author Response · Authors · 2024-11-22
> > **Response to Reviewer hDfK (2/2)**
> >
> > > W2. **Insufficient Benchmarking**: The experiments lack baselines based on protein language models. Specifically, as noted in the original ESM paper, embeddings for protein complexes can be obtained by concatenating the protein chains as input. Given that LLaPA uses ESM embeddings, fine-tuning ESM2 with concatenated inputs is an essential baseline. Additionally, comparisons with ProteinLLM baselines, such as those mentioned in W1, should also be included for a more comprehensive evaluation.
> >
> > Since LLaPA uses ESM-3B as the protein encoder, we fully agree with using ESM-3B as a baseline. It seems that ProLLaMA does not have reported results on the PPI task; while ProtLLM has been tested on PPI, it only involves a simple binary classification task. Given that ProtLLM is also a multimodal large language model for proteins, it is meaningful to include experimental data from this model for reference. We have supplemented the experiments with ESM2-3B and ProtLLM on the mPPI task. For ESM2-3B, we froze its parameters (as our model also has frozen parameters) and trained a multi-classifier on top of it. The experimental results are shown in the table below. As can be seen, ESM2-3B performs poorly when only a multi-classifier is trained with frozen parameters, and ProtLLM shows similar results.
> >
> > |            | SHS27k       | SHS27k        | SHS27k     | SHS148k      | SHS148k       | SHS148k       |
> > | ---------- | ------------ | ------------- | ---------- | ------------ | ------------- | ------------- |
> > |            | random       | dfs           | bfs        | random       | dfs           | bfs           |
> > | DPPI       | 70.45        | 43.69         | 43.87      | 76.10        | 51.43         | 50.80         |
> > | DNN-PPI    | 75.18        | 48.90         | 51.59      | 85.44        | 56.70         | 54.56         |
> > | PIPR       | 79.59        | 52.19         | 47.13      | 88.81        | 61.38         | 58.57         |
> > | ESM2-3B    | 47.58        | 42.50         | 41.97      | 48.92        | 43.06         | 41.25         |
> > | ProtLLM    | 48.67        | 42.77         | 41.94      | 49.29        | 42.66         | 40.33         |
> > | GNN-PPI/R  | 40.53        | 43.19         | 42.52      | 39.48        | 40.96         | 41.42         |
> > | HIGH-PPI/R | 41.51        | 40.06         | 39.87      | 42.81        | 51.06         | 45.94         |
> > | MAPE-PPI/R | 76.84        | 51.69         | 55.21      | 85.96        | 62.13         | 56.68         |
> > | LLaPA      | 82.49(+2.90) | 69.54(+17.35) | 67.21(+12) | 91.78(+2.97) | 73.93(+11.80) | 70.90(+12.33) |
> >
> > > W3. **Redundant Challenges**: Challenges 1 and 2 overlap, both emphasizing issues in graph-based inference at test phase, reducing clarity in presenting the model’s motivation.
> >
> > Challenges 1 and 2 are two distinct challenges with different implications. Challenge (1) refers to the fact that existing methods overlook practical application scenarios when utilizing the PPI network, introducing connection information of the tested protein pairs during testing. Challenge (2) refers to handling completely unseen proteins. Addressing Challenge (1) may introduce new challenges, potentially leading to unseen proteins in specific situations. While this may overlap with Challenge (2) in terms of phenomena, the two are different in their significance.
> >
> > We hope the above response can alleviate your concerns and further endorse our work.

---

> > > ### Comment · Reviewer_hDfK · 2024-11-25
> > > **Response to Rebuttal**
> > >
> > > Thank you for the effort put into the rebuttal.
> > >
> > > However, my primary concerns remain unresolved after reviewing the discussion on ProtLLM and ProLlama. First, regarding the second challenge, the authors claim that these ProtLLMs cannot address the issue of handling unseen proteins, but in fact, they can. Specifically, the protein encoders in these models are inherently capable of generalizing to new proteins without relying on graph-RAG. Second, concerning the third challenge, as acknowledged by the authors, ProtLLM can process multiple proteins together, and ProLlama can extend its context length to handle similar scenarios effectively.
> > >
> > > That said, while I acknowledge the validity of the first challenge, the other two challenges have already been addressed by existing works. As a result, the contributions of this paper appear to be overstated. I believe the paper requires significant revision, with a more comprehensive and critical discussion of these two works.
> > >
> > > Therefore, I intend to maintain a negative score.
> > >
> > > Besides, what are the settings for experiments ProtLLM in the new experiments provided? Is it fine-tuned for mPPI task?

---

> > > > ### Author Response · Authors · 2024-11-26
> > > >
> > > > Thank you for your prompt reply. We understand your concerns and would like to clarify the following points:
> > > >
> > > > **Handling unseen proteins**: Being able to input and encode unseen proteins does not equate to effectively handling them, at least for the mPPI task. The rich topological information provided by the PPI network is indispensable. In our ablation study, we added an experiment without using any PPI network, and the performance of the model was similar to the original ESM2-3B. Moreover, the larger the scale of the PPI network, the more information it can provide. The improvement brought by using UPPIN compared to the original SHS27k network in the ablation experiment also proves this point.
> > > >
> > > > | Pretrain | PPIs Network | LinfoNCE | F1    |
> > > > | -------- | ------------ | -------- | ----- |
> > > > | ✔        |              | ✔        | 43.31 |
> > > > |          | UNI          | ✔        | 62.20 |
> > > > | ✔        | OR           | ✔        | 66.92 |
> > > > | ✔        | UNI          |          | 66.35 |
> > > > | ✔        | UNI          | ✔        | 69.54 |
> > > >
> > > > **Number of input proteins**: We acknowledge that ProtLLM has the ability to input multiple proteins; however, ProLlama's ability in this regard depends on the context length. Even if its context length is extended, it is still very difficult for complex complexes, such as PDB:3ZIA. In addition, ProLlama treats proteins and text as the same modality, which is quite different from LLaPA and ProtLLM. We will clarify in the revised version that LLaPA is not the first work capable of handling any number of proteins.
> > > >
> > > > **Experimental setup of ProtLLM**: ProtLLM itself has not reported any experimental results related to the mPPI task, and its publicly available code does not support mPPI. We downloaded its source code from [ProtLLM](https://github.com/ProtLLM/ProtLLM), its pre-trained weights from [ProtLLM Datasets at Hugging Face](https://huggingface.co/datasets/ProtLLM/ProtLLM), and the weights of ProtST-ESM-2 from [ProtST](https://github.com/DeepGraphLearning/ProtST). We modified the *Ppi4ProtLlmDataset* class and *ProtLlmForBinaryCls* in the code to support the mPPI task, and finally used [ProtLLM/scripts/finetune.sh](https://github.com/ProtLLM/ProtLLM/blob/main/scripts/finetune.sh) for fine-tuning. We did not modify any hyperparameters in scripts/finetune.sh and used the model weights obtained in the 15th epoch (same as LLaPA) for testing.
> > > >
> > > > **Contributions**: Let's put aside the point of 'Number of input proteins' for a moment. We first identified and addressed a widely overlooked issue in the mPPI task; we were the first to effectively incorporate protein features and their corresponding PPI network topological information into LLMs; and our experimental results are also very impressive and comprehensive.
> > > >
> > > > Thank you very much for your review, the time and effort you've put in, and the valuable questions you've raised, I hope these responses can alleviate your concerns. If you have any other questions, feel free to ask, and we will try to respond as time permits. Since the discussion phase has been extended, we may be able to have a more thorough discussion and provide additional supplements.

---

> > > > > ### Comment · Reviewer_hDfK · 2024-11-26
> > > > > **Response to the Authors' Response**
> > > > >
> > > > > Thanks for the prompt feedback.
> > > > >
> > > > > Your new response has convinced me of the effectiveness of the proposed method. However, as I highlighted in my previous comment, two out of the three major challenges have already been addressed by existing methods. Therefore, the paper requires significant revision before it is ready for publication.
> > > > >
> > > > > I do not have any additional experiments to request. However, since the authors still have an opportunity to update the manuscript, it would be more convincing to incorporate these clarifications into the current version of the paper, rather than stating, "we will clarify in the revised version."
> > > > >
> > > > > I do appreciate the substantial effort of the authors during the rebuttal session. Thanks again.

---

> ### Author Response · Authors · 2024-11-27
>
> Dear Reviewer, we have submitted the rebuttal version of our manuscript. For ease of reading, we have marked the revised content in red and used strikethroughs for the deleted content. We have temporarily moved Figure 3 and Figure 5 to the appendix to create more space.
>
> We greatly appreciate the valuable time and effort you have invested in evaluating this work, as well as the insightful questions and suggestions you have provided. We have addressed all of your concerns, and in response to your queries, the following revisions have been made in the manuscript:
>
> 1. We have modified some qualifiers in the introduction to clarify which works the stated limitations are referring to;
> 2. We have added brief descriptions of ProtLLM, ProLLama, Prot2Text, and ProteinGPT in the introduction, and clarified that LLaPA is not the first work to address Challenge 3;
> 3. We have added three new baselines, including ESM2-3B(fixed), ESM2-3B(ft), and ProtLLM;
>
> We hope that all the responses and experimental data provided can alleviate your concerns about this work and further support our efforts.
>
> As there are still a few days left until the end of the discussion, if you have any other questions or concerns, please feel free to raise them, and we will do our utmost to address them.

---

> > ### Comment · Reviewer_hDfK · 2024-11-27
> > **Response to Authors' Rebuttal**
> >
> > I appreciate the authors' considerable effort during the rebuttal session. The updated manuscript has effectively addressed my concerns about the lack of discussion and comparison with related work. As a result, I have adjusted my score accordingly.

---

### Official Review · Reviewer_pBue · 2024-11-03

**Soundness:** 3
**Presentation:** 3
**Contribution:** 3
**Rating:** 6
**Confidence:** 3

**Summary:**

In this paper, the authors proposed a novel method LLaPA (Large Language and Protein Assistant), which integrate protein sequences and PPI networks to improve the protein-protein interaction prediction. The two-stage training strategy enables the model to project the sequence and network information within the LLM representation space. Moreover, the LLaPA can predict the interaction among flexible number of proteins. Extensive experimental results have verified the effectiveness of LLaPA.

**Strengths:**

1.	The authors offer a rational multimodal LLM for protein-protein interaction prediction. In addition, the method can accept more than two proteins to infer the potential relationship among them. It overcomes the evaluation issues present in previous work, demonstrating greater generalizability.

2.	The authors propose a novel knowledge injection method by treating the PPI network as external knowledge and integrating it into the large language model. This approach effectively leverages information from the PPI network, showing promising performance, especially in the prediction of unknown proteins.

3.	It achieved the SOTA performance on the mPPI task and demonstrates significant accuracy in multi-sequence affinity prediction.

**Weaknesses:**

1.	A comparison with some large model-based methods is lacking.

2.	Although the SHS27k and SHS148k datasets are widely used, adding other real biological datasets, such as protein interaction data from different species, would make the experimental results more generalizable and convincing.

3.	The experiments on MA prediction lack persuasiveness and need to be tested on a larger dataset.

**Questions:**

1.	Please describe in detail why the training is divided into two phases, and why there is a need for the first phase. It is suggested that the authors could add an experiment to illustrate the need of the first training phase.

2.    In the prompt, the authors included text representations of the protein sequences and indexes. Please demonstrate the reason.

3.   For training, there are 2 L_LLM. Please demonstrate their difference.

4.   Why do the authors use SGC as graph encoder? What happens if you use gnn like GNN-PPI, e.g. GCN to encode networks? Compared to SGC, it seems like GCN just have one extra step of using a nonlinear activation function.

5.   For the similarity of proteins, the authors used the cosine similarity between protein features. Why not consider sequence alignment scores and structural similarity? Please conduct additional experiments as a supplement.

6.   For the training step 1, the authors pretrained the model on UniprotQA. Is there a possibility of data leakage in this method?

7.  For MA prediction, the authors require the model to directly generate logKd scores. In this setup, the model is likely to mimic training data rather than output genuine scores. The poor performance of Group 6 may be due to its significant divergence from the training data distribution, and analysis shows that the model generated two instances of 7.52 for this group, which could be a high-frequency value in the training data.

8.  The authors have constructed UPPIN by merging STRING, PDBBind, and SAbDab. As the authors state, the STRING contains different types of interactions between proteins, and the other two databases have edges of different types from it, how do the authors handle this here? Could this lead to an inconsistency in the information of the edges in UPPIN, causing the graph encoder to fail?

9. There are over 59 million proteins in the latest release of STRING, so why were only 15,202 proteins considered? And the type of edges in STRING doesn't match with what is mentioned in the article, could the authors please explain it?

10.	In LLM loss (L_LLM), P(*|*;p) should be P(*|*;Q,p).

---

> ### Author Response · Authors · 2024-11-22
> **Response to Reviewer pBue - Part (1/3)**
>
> Thank you for your partial recognition of our work, we also understand your concerns, and we will respond to your comments one by one.
>
> > Please describe in detail why the training is divided into two phases, and why there is a need for the first phase. It is suggested that the authors could add an experiment to illustrate the need of the first training phase.
>
> Two-phase training is adopted by many multimodal large language models [1-5]. The primary purpose of the first phase is to use a large amount of task-agnostic data to align the feature space of the encoder output of other modalities (in this work, proteins) with the input feature space of the large natural language model. The second phase of training is fine-tuning for downstream tasks. The first phase of training is very important and can significantly improve the model's performance, as demonstrated by works [1-5]. Additionally, in our 5.3 ABLATION STUDY, we conducted experiments showing that without the first phase of training, the F1 score decreased by 7.34 on the SHS27k dataset under the DFS setting.
>
> [1] Liu, Haotian, et al. "Visual instruction tuning." *Advances in neural information processing systems* 36 (2024).
>
> [2] Li, Chunyuan, et al. "Llava-med: Training a large language-and-vision assistant for biomedicine in one day." *Advances in Neural Information Processing Systems* 36 (2024).
>
> [3] Chen, Jun, et al. "Minigpt-v2: large language model as a unified interface for vision-language multi-task learning." *arXiv preprint arXiv:2310.09478* (2023).
>
> [4] Liu, Zhiyuan, et al. "MolCA: Molecular Graph-Language Modeling with Cross-Modal Projector and Uni-Modal Adapter." *Proceedings of the 2023 Conference on Empirical Methods in Natural Language Processing*. 2023.
>
> [5] Zhang, Mengmei, et al. "GraphTranslator: Aligning Graph Model to Large Language Model for Open-ended Tasks." *Proceedings of the ACM on Web Conference 2024*. 2024.
>
>
>
> > In the prompt, the authors included text representations of the protein sequences and indexes. Please demonstrate the reason.
>
> The input consists of three parts: the prompt, the protein sequences, and the position of each protein sequence in the protein-protein interaction network (UPPIN). The prompt is encoded by the LLM's encoding layer, the protein sequences are encoded by ESM2, and the protein indexes are used to find the corresponding proteins' positions in UPPIN and obtain the encodings of these nodes in UPPIN. Finally, the protein encodings and node encodings are fused with the prompt encoding at the positions of special tokens in the prompt, forming the final input. We apologize for any confusion caused by the representation in Figure 3, and we will add more detailed explanations in the caption to clarify this.
>
>
>
> > For training, there are 2 L_LLM. Please demonstrate their difference.
>
> Are you referring to the two LLM losses in Figure 1? These two LLM losses correspond to two different training stages. As mentioned in the first question, we conducted an ablation study on the first training stage in section 5.3. The LLM loss in the second stage is essential; without it, the model cannot converge.
>
>
>
> > Why do the authors use SGC as graph encoder? What happens if you use gnn like GNN-PPI, e.g. GCN to encode networks? Compared to SGC, it seems like GCN just have one extra step of using a nonlinear activation function.
>
> We initially used GNN-PPI. In fact, GNN-PPI as an encoder can provide better performance. However, GNN-PPI has higher complexity, requiring more memory and longer training time, especially after we constructed a larger PPI network. We hope to build a more general PPI network in future work, which may be even larger than the current UPPIN. Therefore, we aim to make some trade-offs between performance and efficiency. GCN is the most classic graph encoder, and we considered this method as well. However, after research, we found that SGC can achieve a better balance between performance and efficiency, so we chose SGC.

---

> > ### Author Response · Authors · 2024-11-22
> > **Response to Reviewer pBue - Part (2/3)**
> >
> > > For the similarity of proteins, the authors used the cosine similarity between protein features. Why not consider sequence alignment scores and structural similarity? Please conduct additional experiments as a supplement.
> >
> > We initialize the node features of UPPIN using ESM2. Then, based on the cosine similarity between the target protein and each protein feature in UPPIN, we select the most similar node to obtain its topological information. Protein language models (pLM) like ESM2 can capture sequence correlations and are also trained on structural protein data, allowing them to retain some structural information. Therefore, to some extent, using cosine similarity after encoding with ESM2 encompasses sequence similarity and structural similarity. Several studies [6] [7] have demonstrated that using pLM to obtain protein features and calculate similarity (e.g., cosine similarity, dot product, etc.) is efficient and feasible. If you need any additional supporting materials, please specify the type of experiment you would like, and we will do our utmost to conduct it.
> >
> > [6] Kaminski, Kamil, et al. "pLM-BLAST: distant homology detection based on direct comparison of sequence representations from protein language models." *Bioinformatics* 39.10 (2023): btad579.
> >
> > [7] Liu, Wei, et al. "PLMSearch: Protein language model powers accurate and fast sequence search for remote homology." *Nature communications* 15.1 (2024): 2775.
> >
> >
> >
> > > For the training step 1, the authors pretrained the model on UniprotQA. Is there a possibility of data leakage in this method?
> >
> > We have considered this issue. Specifically, UniprotQA has 1,593 duplicate sequences with SHS27k and 4,781 duplicate sequences with SHS148k. We removed the QA data for these sequences during pre-training, so there is no data leakage. The only potential source of data leakage is the protein encoder ESM-3B, as it was trained on UR50, which includes 48 million protein sequences. We added ESM2-3B as a control group, fixed its parameters, and trained a multi-classifier for experiments on the Multi-label PPI type prediction (mPPI) task. The results show that without fine-tuning the parameters of ESM2-3B, the model struggles to achieve ideal performance. In contrast to the superior performance of LLaPA, this suggests that any potential bias from data leakage in ESM2-3B is minimal.
> >
> > |            | SHS27k       | SHS27k        | SHS27k     | SHS148k      | SHS148k       | SHS148k       |
> > | ---------- | ------------ | ------------- | ---------- | ------------ | ------------- | ------------- |
> > |            | random       | dfs           | bfs        | random       | dfs           | bfs           |
> > | DPPI       | 70.45        | 43.69         | 43.87      | 76.10        | 51.43         | 50.80         |
> > | DNN-PPI    | 75.18        | 48.90         | 51.59      | 85.44        | 56.70         | 54.56         |
> > | PIPR       | 79.59        | 52.19         | 47.13      | 88.81        | 61.38         | 58.57         |
> > | ESM2-3B    | 47.58        | 42.50         | 41.97      | 48.92        | 43.06         | 41.25         |
> > | ProtLLM    | 48.67        | 42.77         | 41.94      | 49.29        | 42.66         | 40.33         |
> > | GNN-PPI/R  | 40.53        | 43.19         | 42.52      | 39.48        | 40.96         | 41.42         |
> > | HIGH-PPI/R | 41.51        | 40.06         | 39.87      | 42.81        | 51.06         | 45.94         |
> > | MAPE-PPI/R | 76.84        | 51.69         | 55.21      | 85.96        | 62.13         | 56.68         |
> > | LLaPA      | 82.49(+2.90) | 69.54(+17.35) | 67.21(+12) | 91.78(+2.97) | 73.93(+11.80) | 70.90(+12.33) |
> >
> >
> >
> > > For MA prediction, the authors require the model to directly generate logKd scores. In this setup, the model is likely to mimic training data rather than output genuine scores. The poor performance of Group 6 may be due to its significant divergence from the training data distribution, and analysis shows that the model generated two instances of 7.52 for this group, which could be a high-frequency value in the training data.
> >
> > As shown in the table below, the value 7.52 appears 12 times in the training set, with a frequency of 1.83%. We have listed information for some values with higher frequencies than 7.52. We hope this data can demonstrate that the two predictions of 7.52 in group 6 are not due to the high frequency of 7.52 in the training set. Additionally, we set up four comparison methods, namely PIPR, E(2), E(3), and E(4). The experimental results are shown in Table 3 of the manuscript, where LLAPA consistently achieved the lowest MAE and the highest PCC.
> >
> > | Value     | 7.52  | 5     | 5.52  | 6     | 6.7   | 7     | 8.7   | 9     |
> > | --------- | ----- | ----- | ----- | ----- | ----- | ----- | ----- | ----- |
> > | Times     | 12    | 16    | 16    | 16    | 17    | 15    | 17    | 32    |
> > | Frequency | 1.83% | 2.44% | 2.44% | 2.44% | 2.59% | 2.29% | 2.59% | 4.88% |

---

> > > ### Author Response · Authors · 2024-11-22
> > > **Response to Reviewer pBue - Part (3/3)**
> > >
> > > > The authors have constructed UPPIN by merging STRING, PDBBind, and SAbDab. As the authors state, the STRING contains different types of interactions between proteins, and the other two databases have edges of different types from it, how do the authors handle this here? Could this lead to an inconsistency in the information of the edges in UPPIN, causing the graph encoder to fail?
> > >
> > > As we mentioned in section 3.4.1, "For UPPIN, we retained all nodes and edges from STRING but removed the edge labels." Our goal is to construct a network of connections between proteins, focusing solely on whether there is a relationship between proteins in this network, without considering the types of edges. Therefore, we do not encounter issues of "inconsistency in the information of the edges" or encoding failures.
> > >
> > >
> > >
> > > > There are over 59 million proteins in the latest release of STRING, so why were only 15,202 proteins considered? And the type of edges in STRING doesn't match with what is mentioned in the article, could the authors please explain it?
> > >
> > > Like works [8-10], we used the Homo sapiens subset from STRING, which may have more research value. Thank you for pointing out this concern; we will clarify this in the revised version.
> > >
> > > [8] Chen, Muhao, et al. "Multifaceted protein–protein interaction prediction based on Siamese residual RCNN." *Bioinformatics* 35.14 (2019): i305-i314.
> > >
> > > [9] Lv, Guofeng, et al. "Learning unknown from correlations: Graph neural network for inter-novel-protein interaction prediction." *arXiv preprint arXiv:2105.06709* (2021).
> > >
> > > [10] Wu, Lirong, et al. "MAPE-PPI: Towards Effective and Efficient Protein-Protein Interaction Prediction via Microenvironment-Aware Protein Embedding." *The Twelfth International Conference on Learning Representations*.
> > >
> > >
> > >
> > > > In LLM loss (L_LLM), P(|;p) should be P(|;Q,p).
> > >
> > > Thank you for pointing out this issue; we will correct it in the revised version.
> > >
> > > We hope the above response can alleviate your concerns and further endorse our work.

---

> ### Author Response · Authors · 2024-11-27
>
> Dear Reviewer, we have submitted the rebuttal version of our manuscript. For ease of reading, we have marked the revised content in red and used strikethroughs for the deleted content. We have temporarily moved Figure 3 and Figure 5 to the appendix to create more space.
>
> We greatly appreciate the valuable time and effort you have invested in evaluating this work, as well as the insightful questions you have raised. We have answered all of your concerns, and your queries have led to the following three revisions in the manuscript:
>
> 1. Added a more detailed explanation for Figure 3;
> 2. Explicitly stated that the Homo sapiens subset of the STRING dataset was used;
> 3. Corrected the formula for LLM loss;
>
> Additionally, we have included more related works on protein-LLMs; added ESM2-3B (fixed weights), ESM2-3B (fine-tuned), and ProtLLM as baselines; clarified the definition of the MA task; and provided MA experimental results considering all sequences rather than unique sequences (Appendix A.10).
>
> We hope that all the responses provided can alleviate your concerns about this work and further support our efforts.
>
> As there are still a few days left until the end of the discussion, if you have any other questions or concerns, please feel free to raise them, and we will do our utmost to address them.

---

> ### Author Response · Authors · 2024-11-29
>
> Dear Reviewer pBue,
>
> Thank you for the time and effort you have dedicated to evaluating our work.
>
> We have responded to your questions point by point, but we have not yet received your feedback. We are very eager to know if we have addressed your concerns regarding this work.
>
> Although there are still three days left for the discussion, the weekend is approaching, which might be your personal time. Therefore, the actual time available for discussion may be limited.
>
> I sincerely hope to receive your response to let us know if we have resolved your concerns.
>
> Additionally, if you require any extra supporting materials, we still have time to prepare them.
>
> Thank you once again, and we look forward to your reply.

---

> > ### Comment · Reviewer_pBue · 2024-12-03
> >
> > Thanks for the responses. I have raised the score accordingly.

---

### Official Review · Reviewer_waWJ · 2024-11-05

**Soundness:** 2
**Presentation:** 3
**Contribution:** 2
**Rating:** 5
**Confidence:** 5

**Summary:**

The paper introduces LLaPA, a multimodal large language model that integrates PPI networks as external knowledge to predict protein-protein interactions and affinity.

**Strengths:**

1. **Writing**. The writing is mostly clear and easy to follow.
2. **Significance**. Predicting protein-protein interactions and affinity is an important problem in biology and medicine. Comparing models in a realistic setting is a good practice.

**Weaknesses:**

1. **Potential major error in chemistry facts**. The Multi-sequence Affinity Prediction is theoretically not a well-defined task. For antibody chains H+L and antigen chain A, the antibody-antigen binding affinity (the usually measured one) is defined by $K_\text{D} = \frac{[\text{HL}][\text{A}]}{[\text{HLA}]}$, i.e. between the entity HL (the antibody) and A (the antigen). Usually, for a set of chains, a single affinity value does not exist; rather, it is dependent on the choice of two binding partners.
2. **Unclear function of LLM**. The LLM in this paper acts as a vessel to integrate the PPI network and the protein embedding. Since the tasks you study are closed-domain (i.e. no open answer needed), you could theoretically replace it with a smaller network, e.g. Graph Transformer, that inputs the target proteins and their related PPI graph embeddings (the cosine similarity between the target protein and the related protein on the PPI graph can be incorporated into this graph embedding), and outputs the mPPI logits or logKd values. It is unclear how much benefit the LLM brings to the task.
3. **Unfair comparison**. In the main text, the authors present results for the "/R" models where the test edges are removed. As these models are trained on the PPI graph, it is to be expected that the "/R" models will underperform significantly due to a significant gap in training and test settings. Outperforming these nerfed models is not a convincing achievement of LLaPA.

**Questions:**

1. Regarding weakness #2, could you provide more theoretical or emperical evidence to justify the benefit of LLM?
2. Regarding weakness #3, if removing test edges makes the graph structure useless, the authors should add one or more PPI-graph-free baselines for comparison, e.g. ESM2/3 or SaProt. The protein encoders can be directly trained on the train edges and tested on the test edges.

---

> ### Author Response · Authors · 2024-11-22
> **Response to Reviewer waWJ - Part (1/2)**
>
> Thank you for your partial recognition of our work. Your concerns are also valid, and we will address your questions one by one.
>
> > Usually, for a set of chains, a single affinity value does not exist; rather, it is dependent on the choice of two binding partners.
>
> According to the definition of the dissociation constant, the affinity of a protein complex depends on the choice of the binding partner. There are indeed some ambiguities when defining the MA task. The correct definition of the MA task should be: given a complex $C=(B,T)$，where $B$ refers to the binder and $T$ refers to the target, predict its logarithmic dissociation constant $\log{Kd}=\log\frac{[B][T]}{[BT]}$. $B$ and $T$ can each be a single protein sequence or a complex containing multiple sequences. For example, in PDB 6ILM, the affinity describes the dissociation constant between the Fc receptor and Echovirus 6. However, the target Echovirus 6 is a complex containing four protein sequences. For the PDB2020 dataset, which includes 2852 complexes, accurately extracting the binder and target based on the given information is very challenging and requires manual analysis of the papers corresponding to each PDB entry. Therefore, we have simplified this task in the current version.
>
> > Regarding weakness #2, could you provide more theoretical or emperical evidence to justify the benefit of LLM?
>
> You are absolutely right. Theoretically, LLM is not a necessity. For both the Multi-label PPI type prediction task and the Multi-sequence Affinity prediction task, we could use a smaller network architecture to achieve these tasks individually. LLM is one solution to the issues we raised, but it is not the only solution. However, using LLM has unique advantages:
>
> 1. LLM possesses excellent multitasking capabilities and flexibility. By constructing appropriate natural language instructions, we can unify various tasks into a generative task, using a unified framework and training method to build a multi-task unified model. Different downstream tasks can be guided through appropriate natural language instructions.
>
> 2. Additionally, small models fit different loss functions for different tasks, and the model itself does not know what it is doing. In contrast, large language models have a certain level of common sense and seem to have the ability to understand tasks. This means that LLM might know what task it is performing, what the task means, and even the hidden implications of the output. For example, if we ask gpt-4o-mini, "If the relationship between protein A and protein B is activation, what does it mean?", we would get: "If the relationship between protein A and protein B is described as 'activation,' it means that protein A enhances or stimulates the activity, function, or expression of protein B. This can involve various mechanisms, such as promoting a conformational change in protein B, facilitating its interaction with other molecules, or increasing its stability or availability within the cellular environment. In essence, protein A acts as a positive regulator of protein B's biological activity."
> 3. The LLMs demonstrates excellent few-shot learning and zero-shot learning capabilities [1-2]. By connecting the protein modality with the LLMs, it may enable the LLMs to understand proteins and provide a richer array of application scenarios.
>
> More specific discussions can be found in references [1-6].
>
> In this work, we started from specific practical tasks and hope to build a unified, practically valuable multimodal protein model in the future, which is why we used LLM in this work. However, in terms of performance, whether using LLM will necessarily be more effective than a specially trained small model for specific tasks is indeed a topic worth discussing and requires more comprehensive work for validation.
>
> [1] Kojima, Takeshi, et al. "Large language models are zero-shot reasoners." *Advances in neural information processing systems* 35 (2022): 22199-22213.
>
> [2] Gruver, Nate, et al. "Large language models are zero-shot time series forecasters." *Advances in Neural Information Processing Systems* 36 (2024).
>
> [3] He, Qianyu, et al. "Can Large Language Models Understand Real-World Complex Instructions?." *Proceedings of the AAAI Conference on Artificial Intelligence*. Vol. 38. No. 16. 2024.
>
> [4] Yuan, Quan, et al. "Tasklama: probing the complex task understanding of language models." *Proceedings of the AAAI Conference on Artificial Intelligence*. Vol. 38. No. 17. 2024.
>
> [5] Zhao, Zirui, Wee Sun Lee, and David Hsu. "Large language models as commonsense knowledge for large-scale task planning." *Advances in Neural Information Processing Systems* 36 (2024).
>
> [6] Guo, Taicheng, et al. "What can large language models do in chemistry? a comprehensive benchmark on eight tasks." *Advances in Neural Information Processing Systems* 36 (2023): 59662-59688.

---

> ### Author Response · Authors · 2024-11-22
> **Response to Reviewer waWJ - Part (2/2)**
>
> > In the main text, the authors present results for the "/R" models where the test edges are removed. As these models are trained on the PPI graph, it is to be expected that the "/R" models will underperform significantly due to a significant gap in training and test settings. Outperforming these nerfed models is not a convincing achievement of LLaPA.
>
> We understand your concern that this comparison might seem unfair. It is important to note that a key focus of our work is to push PPI prediction towards a more practical direction. Therefore, we included the '/R' setting in the comparison group, as not including '/R' would also be unfair. We can see that starting from GNN-PPI, methods based on PPI networks have shown a significant leap in performance on the multi-label PPI type prediction task. However, this improvement is inherently unfair because these methods have potential data leakage during testing. Therefore, we believe that removing test edges might provide a fairer comparison. Although removing the edges between test proteins only during testing unsurprisingly reduces the model's predictive ability, removing test edges during training results in even worse performance, especially under more challenging DFS and BFS splits, because a considerable number of isolated nodes cannot obtain any information from the PPI network. Excluding these PPI network-based methods, PIPR might be the best mPPI prediction method currently available. In the revised version, we have added ESM-3B and ProtLLM [8] as baselines. For ESM2-3B, we froze its parameters (as our model also has frozen parameters) and trained a multi-classifier on top of it. The experimental results are shown in the table below. As can be seen, ESM2-3B performs poorly when only a multi-classifier is trained with frozen parameters, and ProtLLM shows similar results.
>
> |            | SHS27k       | SHS27k        | SHS27k     | SHS148k      | SHS148k       | SHS148k       |
> | ---------- | ------------ | ------------- | ---------- | ------------ | ------------- | ------------- |
> |            | random       | dfs           | bfs        | random       | dfs           | bfs           |
> | DPPI       | 70.45        | 43.69         | 43.87      | 76.10        | 51.43         | 50.80         |
> | DNN-PPI    | 75.18        | 48.90         | 51.59      | 85.44        | 56.70         | 54.56         |
> | PIPR       | 79.59        | 52.19         | 47.13      | 88.81        | 61.38         | 58.57         |
> | ESM2-3B    | 47.58        | 42.50         | 41.97      | 48.92        | 43.06         | 41.25         |
> | ProtLLM    | 48.67        | 42.77         | 41.94      | 49.29        | 42.66         | 40.33         |
> | GNN-PPI/R  | 40.53        | 43.19         | 42.52      | 39.48        | 40.96         | 41.42         |
> | HIGH-PPI/R | 41.51        | 40.06         | 39.87      | 42.81        | 51.06         | 45.94         |
> | MAPE-PPI/R | 76.84        | 51.69         | 55.21      | 85.96        | 62.13         | 56.68         |
> | LLaPA      | 82.49(+2.90) | 69.54(+17.35) | 67.21(+12) | 91.78(+2.97) | 73.93(+11.80) | 70.90(+12.33) |
>
> We hope the above response can alleviate your concerns and further endorse our work.

---

> > ### Comment · Reviewer_waWJ · 2024-11-26
> > **Response to the authors**
> >
> > I appreciate the time and effort you have made in responding and revising the paper. After carefully reviewing your response and other reviewers' comments, I have decided to maintain my score of 5.
> >
> > Specifically, the training method of ESM2-3B is not very convincing, as only one linear layer is used for a complex task like PPI. I expect a significant performance increase if either (1) multiple transformer layers were trained after a fixed encoder, or (2) the model is fine-tuned.

---

> > > ### Author Response · Authors · 2024-11-26
> > >
> > > Thank you for your response and the time you have dedicated.
> > >
> > > We understand your concerns. With the extended discussion period, we will have time to conduct additional experiments. Since your primary concern is whether full parameter fine-tuning of ESM2-3B can achieve better performance, we will redesign a set of comparative experiments to fully fine-tune ESM2-3B, in addition to the previous approach of fine-tuning only the multi-classifier. This is not difficult; fully fine-tuning ESM2-3B for 15 epochs (similar to LLaPA) on 8 A100 GPUs takes only about one hour. It is worth mentioning that fixing the parameters of ESM2-3B is also reasonable, as LLaPA does not train its parameters. If you have any other concerns, please feel free to point them out, and we will do our best to address them.

---

> > > ### Author Response · Authors · 2024-11-29
> > >
> > > Dear Reviewer waWJ,
> > >
> > > Thank you for the time and effort you have dedicated to evaluating our work.
> > >
> > > Your main concern was that using ESM2-3B with fixed parameters as a baseline was not convincing enough. Therefore, we conducted new experiments with full parameter fine-tuning of ESM2-3B, referred to as ESM2-3B(ft). The results show that ESM2-3B(ft) provides a stronger baseline on SHS27k (dfs), SHS148k (dfs), and SHS148k (bfs), yet it still does not surpass our proposed LLaPA. All the revisions have been incorporated into the rebuttal version of the manuscript.
> > >
> > > We aim to avoid using terms like 'first' or 'pioneer' to exaggerate our work. However, we would like to emphasize that we are the first to identify and address the improper use of PPI networks in existing methods. We are also the first to integrate both protein features and PPI network features into LLMs and demonstrate their effectiveness.
> > >
> > > We hope these new experimental results will earn your greater support, which is very important to us. As we are still in the discussion phase, we have more opportunities to address and resolve your concerns.
> > >
> > > Thank you once again.

---

> ### Author Response · Authors · 2024-11-27
>
> We greatly appreciate the time and effort you have invested in evaluating this work. We have added a new baseline, which involves full parameter fine-tuning of ESM2-3B. We modified the EsmClassificationHead in EsmForSequenceClassification from the transformers library [transformers/src/transformers/models/esm/modeling_esm.py at v4.46.3 · huggingface/transformers](https://github.com/huggingface/transformers/blob/v4.46.3/src/transformers/models/esm/modeling_esm.py#L1077) to support the mPPI task. For each task, we trained for 15 epochs (same as LLaPA), and the other hyperparameters of the experiment were also consistent with LLaPA. On 8 A100 40G, each SHS27k-related task required about 1 hour of training; each SHS148k-related task required about 5 hours. After full parameter fine-tuning, ESM2-3B provided a stronger baseline on SHS27k (dfs), SHS148k(dfs), and SHS148k(bfs), but it did not perform better in other experimental settings. LLaPA still achieved the best performance under various task settings.
>
> |                 | SHS27k  | SHS27k  | SHS27k | SHS148k | SHS148k | SHS148k |
> | --------------- | ------- | ------- | ------ | ------- | ------- | ------- |
> |                 | random  | dfs     | bfs    | random  | dfs     | bfs     |
> | DPPI            | 70.45   | 43.69   | 43.87  | 76.10   | 51.43   | 50.80   |
> | DNN-PPI         | 75.18   | 48.90   | 51.59  | 85.44   | 56.70   | 54.56   |
> | PIPR            | 79.59   | 52.19   | 47.13  | 88.81   | 61.38   | 58.57   |
> | ESM2-3B (fixed) | 47.58   | 42.50   | 41.97  | 48.92   | 43.06   | 41.25   |
> | ESM2-3B (ft)    | 79.23   | 63.38   | 48.80  | 87.86   | 66.92   | 61.88   |
> | ProtLLM         | 48.67   | 42.77   | 41.94  | 49.29   | 42.66   | 40.33   |
> | GNN-PPI/R       | 40.53   | 43.19   | 42.52  | 39.48   | 40.96   | 41.42   |
> | HIGH-PPI/R      | 41.51   | 40.06   | 39.87  | 42.81   | 51.06   | 45.94   |
> | MAPE-PPI/R      | 76.84   | 51.69   | 55.21  | 85.96   | 62.13   | 56.68   |
> | LLaPA           | 82.49   | 69.54   | 67.21  | 91.78   | 73.93   | 70.90   |
> |                 | (+2.90) | (+6.16) | (+12)  | (+2.97) | (+7.01) | (+9.02) |
>
>
> Additionally, we have listed the ratios of protein pairs tested under each experimental setting that fall into the categories of 'both have been seen' (BS), 'either one protein has been seen' (ES), and 'neither one has been seen' (NS), as shown in the following table:
> |      | SHS27k | SHS27k | SHS27k | SHS148k | SHS148k | SHS148k |
> | ---- | ------ | ------ | ------ | ------- | ------- | ------- |
> |      | random | dfs    | bfs    | random  | dfs     | bfs     |
> | BS   | 90.07% | 0.00%  | 0.00%  | 95.65%  | 0.28%   | 0.00%   |
> | ES   | 9.28%  | 80.13% | 69.54% | 4.26%   | 85.62%  | 78.10%  |
> | NS   | 0.65%  | 19.87% | 30.46% | 0.09%   | 14.10%  | 21.90%  |
>
> We hope that the addition of this new baseline can further substantiate the effectiveness of LLaPA and alleviate your concerns. We will update these experimental results in the rebuttal version of the manuscript and submit it before the deadline. If you have any other questions or concerns, please feel free to raise them. As there are still a few days left before the discussion deadline, we have the opportunity for more in-depth discussions.

---

> > ### Author Response · Authors · 2024-11-27
> >
> > Dear Reviewer, we have submitted the rebuttal version of the manuscript. For ease of reading, we have marked the modified content in red and the deleted content with strikethroughs. We have temporarily moved Figure 3 and Figure 5 to the appendix to free up more space.
> >
> > We greatly appreciate the valuable time and effort you have put into evaluating this work, as well as the valuable questions and suggestions you have raised. We have addressed all of your concerns, and in response to your questions, the following two points are reflected in the revised version:
> >
> > 1. A clearer definition of the MA task;
> > 2. More baselines, including ESM2-3B (fixed), ESM2-3B (ft), and ProtLLM.
> >
> > We hope that all the responses and experimental data provided can alleviate your concerns about this work and further support our efforts.
> >
> > As there are still a few days left until the end of the discussion, if you have any other questions or concerns, please feel free to raise them, and we will do our utmost to address them.

---

### Note · Authors · 2025-05-26

I have read and agree with the venue's withdrawal policy on behalf of myself and my co-authors.

---

### Meta-Review · Area_Chair_NQor · 2024-12-21

**Metareview:**

This paper introduces a multimodal large language model (LLM), called LLaPA (Large Language and Protein Assistant), which aims to offer a novel approach that improves the integration of information regarding proteins and PPI networks to improve PPI prediction performance over other existing mostly semi-supervised PPI prediction schemes.
Overall, reviewers note that the paper has been written clearly, bringing in multi-modal aspects to enhance PPI prediction is meaningful, and evaluation results in the current manuscript appear promising.
However, there are also concerns about data leakage, novelty and significance of improvement over other SOTA methods, the main motivation and benefits of incorporating LLM over smaller/simpler models (e.g., graph transfomers).
Furthermore, while the current evaluation results on certain PPI tasks are promising, the overall assessment is not yet fully convincing to demonstrate LLaPA's benefits over current SOTA.

**Additional Comments On Reviewer Discussion:**

The authors have very actively engaged in the discussion, and they have provided additional experimental results and clarifications, which have addressed the reviewers' initial doubts and concerns to some extent.
However, some of the major concerns do not appear to have been fully addressed and the manuscript may require a major revision to address them.

---

### Decision · Program_Chairs · 2025-01-22

Reject